# Causal evidence for a domain-specific role of left superior frontal sulcus in human perceptual decision-making

**Miguel Barretto-Garcia[1,2]\*[†], Marcus Grueschow[1†], Marius Moisa[1], Rafael Polania[3], Christian C Ruff[1]**

[1]Zurich Center for Neuroeconomics (ZNE), Department of Economics, University of Zurich, Zurich, Switzerland; [2]Department of Neuroscience, Washington University in St. Louis, St. Louis, United States; [3]Decision Neuroscience Lab, Department of Health Sciences and Technology, ETH Zurich, Zurich, Switzerland

**\*For correspondence:**
gmiguel@wustl.edu

[†]These authors contributed equally to this work

**Competing interest:** The authors declare that no competing interests exist.

## eLife Assessment

In this **important** paper, Garcia et al seek to determine whether the superior frontal sulcus (SFS), an area previously implicated in evidence accumulation for perceptual decisions, plays a causal role in perceptual and/or value-based decisions. Through a combination of careful paradigm design, computational modelling, transcranial magnetic stimulation and fMRI analyses, the authors provide **convincing** evidence that the SFS supports perceptual but not value-based decisions and that its disruption leads to a lowering of decision boundaries.

**Abstract** Humans and animals can flexibly choose their actions based on different information, ranging from objective states of the environment (e.g., apples are bigger than cherries) to subjective preferences (e.g., cherries are tastier than apples). Whether the brain instantiates these different choices by recruiting either specialised or shared neural circuitry remains debated. Specifically, domain-general accounts of prefrontal cortex (PFC) function propose that prefrontal areas flexibly process either perceptual or value-based evidence depending on what is required for the present choice, whereas domain-specific theories posit that PFC sub-areas, such as the left superior frontal sulcus (SFS), selectively integrate evidence relevant for perceptual decisions. Here, we comprehensively test the functional role of the left SFS for choices based on perceptual- and value-based evidence, by combining functional magnetic resonance imaging with a behavioural paradigm, computational modelling, and transcranial magnetic stimulation (TMS). Confirming predictions by a sequential sampling model, we show that TMS-induced excitability reduction of the left SFS selectively changes the processing of decision-relevant perceptual information and associated neural processes. In contrast, value-based decision-making and associated neural processes remain unaffected. This specificity of SFS function is evident at all levels of analysis (behavioural, computational, and neural, including functional connectivity), demonstrating that the left SFS causally contributes to evidence integration for perceptual but not value-based decisions.

## Introduction

Humans and animals alike perform a mélange of goal-directed decisions that require the accumulation of different types of information. If the goal, for example, is to accurately determine whether an apple is bigger than a cherry (perceptual choice), the decision-maker accumulates size information of each fruit; or, the decision-maker may draw out information from personal taste profiles if the goal is to

determine whether consuming a cherry over an apple maximises their subjective preferences (value-based choice). Previous studies have shown that different brain circuitries are recruited to accumulate evidence that would instantiate such distinct goal-directed decisions (*Summerfield and Tsetsos, 2012*; *Polanía et al., 2014*; *Polanía et al., 2015*; *Grueschow et al., 2015*); thus, it remains debated to what degree certain decision-making processes share neural circuitry or whether these processes operate under specialised systems. However, prior studies were largely correlational (*Heekeren et al., 2004*; *Heekeren et al., 2008*; *Polanía et al., 2014*; *Grueschow et al., 2015*), and most causal studies were only limited to one type of choice (*Philiastides et al., 2011*; *Rahnev et al., 2016*) and performed in animals (*Ding and Gold, 2012b*; *Erlich et al., 2015*; *Hanks et al., 2015*). Animal studies provide critical causal insights, yet direct translation to humans can be limited by species-specific anatomy and potential non-homologies (e.g., human superior frontal sulcus [SFS] versus frontal orienting fields in rodents). Therefore, establishing causal contributions in the human brain remains essential.

In the absence of a comparison choice task, it is impossible to ascertain whether neural circuitry is domain-specific to a particular process, or domain-general that it may be involved across many types of choices. Very few studies (*Polanía et al., 2014*; *Polanía et al., 2015*; *Grueschow et al., 2015*) have carefully matched perceptual- and value-based decisions in terms of evidence strength, stimulus display, and response modality, and compared them through the lens of a common sequential sampling framework of evidence accumulation (*Dutilh and Rieskamp, 2016*; *Gold and Shadlen, 2007*; *Krajbich, 2019*), which has long been applied to both perceptual- (*Ratcliff and McKoon, 2008*) and value-based (*Busemeyer and Townsend, 1993*; *Usher and McClelland, 2001*) decisions. Such studies were able to identify common and specialised circuitries and mechanisms associated with perceptual- or value-based decisions or both (*Polanía et al., 2014*; *Grueschow et al., 2015*).

But unless causality is established, it is even more difficult to attribute the circuitry's role in evidence accumulation for one or several choice domains, or whether its involvement is peripheral and merely functionally supporting a larger system. Given task complexity, such studies of observing causal neural effects in healthy human populations using non-invasive brain stimulation are incredibly sparse. One previous study has, at least, shown that causally de-synchronising frontoparietal connectivity specifically increased choice variability during value-based choice, but had no effect on perceptual decisions *Polanía et al., 2015*; thus, establishing the causal role of the frontoparietal network during value-based choice. But while indeed causal, the study was limited, relative to the standards of evidence in animal studies (*Erlich et al., 2015*; *Hanks et al., 2015*; *Piet et al., 2017*), since its results, as in many causal stimulation studies in humans (*Philiastides et al., 2011*; *Rahnev et al., 2016*), showed behavioural, but no neural effects. Furthermore, we only have evidence of a single dissociation that shows a causal stimulation effect specific to value-based, and not perceptual choice. What candidate region would show a causal effect that is specific to perceptual, not value-based decisions in a way that would demonstrate a double dissociation?

Seminal human imaging studies have repeatedly implicated the SFS, a posterior portion of the dorsolateral prefrontal cortex (dlPFC), during perceptual decision-making (*Heekeren et al., 2004*; *Heekeren et al., 2008*; *Mulder et al., 2014*). While these studies have shown correlational evidence, it remains challenging to establish whether the SFS is directly involved in evidence accumulation or whether its observed activity reflects upstream or downstream support processes (e.g., attention or working memory maintenance) rather than the accumulation computation per se. In this context, it is conceivable to imagine that the SFS would only play a role in a broader network that is not specific to the evidence accumulation process. Moreover, causal evidence from human studies is difficult to obtain because most prior causal studies were only limited to one type of choice or performed in animals, and it is unclear if findings can be translated from animal models to human decision-making. However, disruption of human left SFS with non-invasive stimulation has been shown to impact behavioural performance and response speed in a dynamic face-house classification task, in a manner consistent with a reduction of evidence accumulation during decision-making. However, the domain-specificity of SFS contribution is unclear. Some studies have shown that dlPFC activity may reflect value-based evidence integration (*Basten et al., 2010*; *Sokol-Hessner et al., 2012*), suggesting the domain-generality of prefrontal function (*Owen, 1997*; *Petrides, 2005*). However, it is hard to directly compare the implicated neural processes to those that underlie perceptual decision-making processes, due to major differences in the stimuli and experimental approaches classically used in each domain (*Gold and Shadlen, 2007*; *Heekeren*

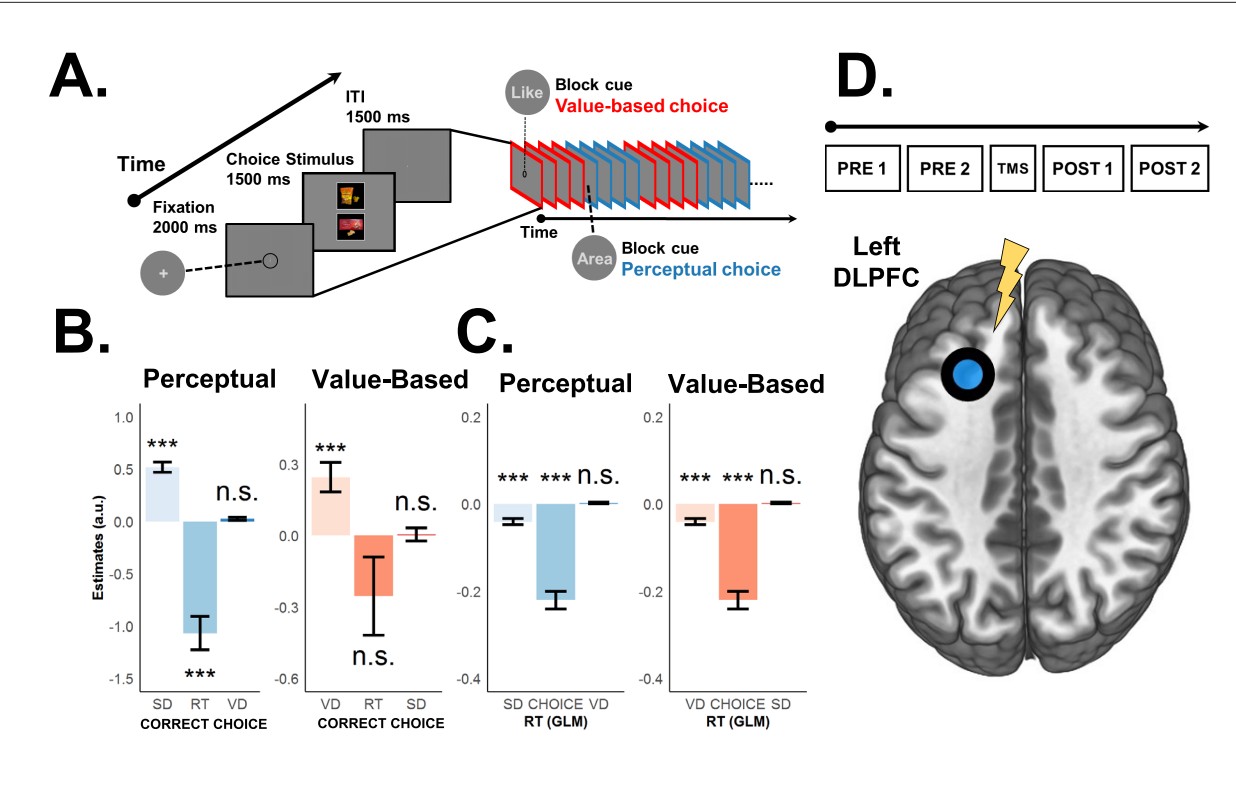

**Figure 1.** Behavioural food choice paradigm, theta-burst stimulation protocol, and behavioural regressions. (**A**) Example of decision stage. Participants were cued in advance about the type of decision required. Perceptual decisions required participants to choose the food item with the largest size while value-based decisions required participants to choose the food item they preferred to consume at the end of the experiment. Participants alternated between blocks of perceptual (blue) or value-based (red) choice trials (7–9 trials per task-block). (**B**) Logistic regression results show that the larger the evidence strength, the more likely decision-makers will respond accurately. Choice accuracy is only related to the evidence that is currently task-relevant (size difference [SD] for perceptual or value difference [VD] for value-based choice), not to the task-irrelevant evidence (RT is reaction time of current choice). (**C**) Similarly, our linear regressions show that RTs are negatively associated only with the task-relevant evidence (and lower for perceptual choices overall, captured by regressor CH (1 = perceptual, 0 = value-based)). Consistent with previous findings, the results in (**B**) and (**C**) confirm that our paradigm can distinguish and compare evidence processing for matched perceptual- and value-based decisions. Error bars in (**B**) and (**C**) represent the 95% confidence interval range of the estimated effect sizes. *$p < 0.05$, **$p < 0.01$, and ***$p < 0.001$. (**D**) Theta-burst stimulation protocol. After the fourth pre-TMS run, participants received continuous theta-burst stimulation (cTBS) over the left superior frontal sulcus (SFS) region of interest (ROI) (area encircled and coloured blue). cTBS consisted of 200 trains of 600 pulses of 5 Hz frequency for 50 s.

The online version of this article includes the following figure supplement(s) for figure 1:

**Figure supplement 1.** Domain-general and domain-specific regions involved in perceptual- and value-based decisions.

et al., 2004), and that direct and principled comparisons with other decision-making domains, in general, are largely missing.

Here, we test the domain-specificity of the left SFS and address the crucial double-dissociation gap in the literature by applying continuous theta-burst transcranial magnetic stimulation (cTBS) followed by functional magnetic resonance imaging (fMRI) while human participants alternated between matched perceptual- and value-based choices (*Polanía et al., 2014*; *Polanía et al., 2015*). We modelled the observed behavioural changes with the DDM, allowing us to causally associate the stimulated SFS region to specific underlying latent subprocesses of the unfolding decision (*Mulder et al., 2014*; *Polanía et al., 2015*) as well as BOLD activation. Thus, this common evidence accumulation framework provides us with clear testable hypotheses regarding possible effect patterns across behavioural, computational, and neural levels.

# Results

## The experiment

We recorded fMRI data from hungry, healthy participants ($n = 20$) performing perceptual- and value-based choice tasks in alternation (Methods and *Figure 1B*). For perceptual decisions, participants chose the larger food item, while for value-based decisions, participants chose the food item that they would preferably receive and consume by the end of the experiment. The stimuli and motor responses were identical for both tasks, as in previous experiments (*Polanía et al., 2014*; *Polanía et al., 2015*). Choice pairings were predetermined based on participants' individual subjective perceptual- and value-based ratings of the food items, obtained just prior to the scanning session. Perceptual evidence was defined as the size difference (SD) between the food items, whereas value evidence was defined as the difference in value ratings (VD) between the choice alternatives (see Methods and *Figure 1B*). A choice was classified as correct when it was consistent with the previously acquired ratings regarding size and preference, respectively, that is, when the larger-rated item was chosen for perceptual decisions or the higher-valued item was chosen for value-based decisions (*Polanía et al., 2014*; *Polanía et al., 2015*). Our sample size is well within acceptable range, similar to that of previous transcranial magnetic stimulation (TMS) studies (*Philiastides et al., 2011*; *Rahnev et al., 2016*; *Jackson et al., 2021*; *van der Plas et al., 2021*; *Murd et al., 2020*).

Our experiment was divided into pre- and post-stimulation blocks. After participants had performed four pre-stimulation session-blocks inside the scanner, they received continuous theta-burst stimulation (cTBS) (*Huang et al., 2005*; *Di Lazzaro et al., 2005*; *Di Lazzaro et al., 2008*) over the left SFS (MNI coordinates, $x = -24, y = 24, z = 36$; *Heekeren et al., 2004*; *Philiastides et al., 2011*; *Grueschow et al., 2018*). Following this intervention, participants completed four post-stimulation fMRI blocks. By comparing the effects of stimulation on both types of behaviour and brain activity between post- and pre-stimulation blocks, we identify the role of SFS for either type of decision-making. In particular, we examined whether the SFS is indeed selectively involved in perceptual decisions as previously suggested (*Heekeren et al., 2004*; *Heekeren et al., 2006*; *Rahnev et al., 2016*; *Philiastides et al., 2011*).

Our initial fMRI analyses were conducted at two levels. The first analysis aimed to broadly identify brain areas recruited for each choice task and those common to both. In this analysis, we assessed the average BOLD activity at the task level (perceptual versus value-based), irrespective of evidence accumulation. The second analysis focused on areas representing evidence accumulation specific to each type of choice. Here, we assessed how BOLD activity is modulated by trial-by-trial evidence strength. For this analysis, we used evidence strength from each choice task (perceptual or value-based) as a parametric modulator and regressed trial-by-trial evidence strength with BOLD (see Methods).

## Study hypotheses

Previous studies have identified the causal mechanistic role of the SFS in evidence accumulation during perceptual decision-making (*Heekeren et al., 2004*; *Heekeren et al., 2006*; *Rahnev et al., 2016*), and its effect on stimulation may arise from one of either two channels. That is, one study reported that SFS disruption during a speeded perceptual categorisation task reduced accuracy and increased response times (*Philiastides et al., 2011*) and found associated decreases in drift rate, the DDM parameter describing the efficiency of sensory evidence integration. In contrast, another human brain stimulation study suggested that behavioural changes due to SFS disruption during a perceptual two-alternative-forced-choice (2AFC) task reflect decreases in the decision threshold, characterised by faster response speed but decreased choice precision. Simulations with the same DDM modelling framework (*Rahnev et al., 2016*) suggested that the decision threshold parameter could account for individual behavioural changes. Simultaneously acquired fMRI data suggested that SFS does not code the rate of integration but rather the necessary amount of evidence to be accumulated for the perceptual choice at hand (*Rahnev et al., 2016*).

To this end, we hypothesise that if SFS neurons indeed selectively accumulate perceptual evidence, we should find that their inhibition by cTBS leads to decreases in choice precision and increases in reaction times, a behavioural pattern that corresponds to a decrease in the DDM drift-rate parameter, and to concurrent increases in BOLD signals (caused by prolonged neural evidence accumulation; *Figure 2A–C*). Critically, a different pattern can be expected when SFS neurons are involved in setting the criterion, i.e., determining the amount of evidence that needs to be accumulated for a perceptual

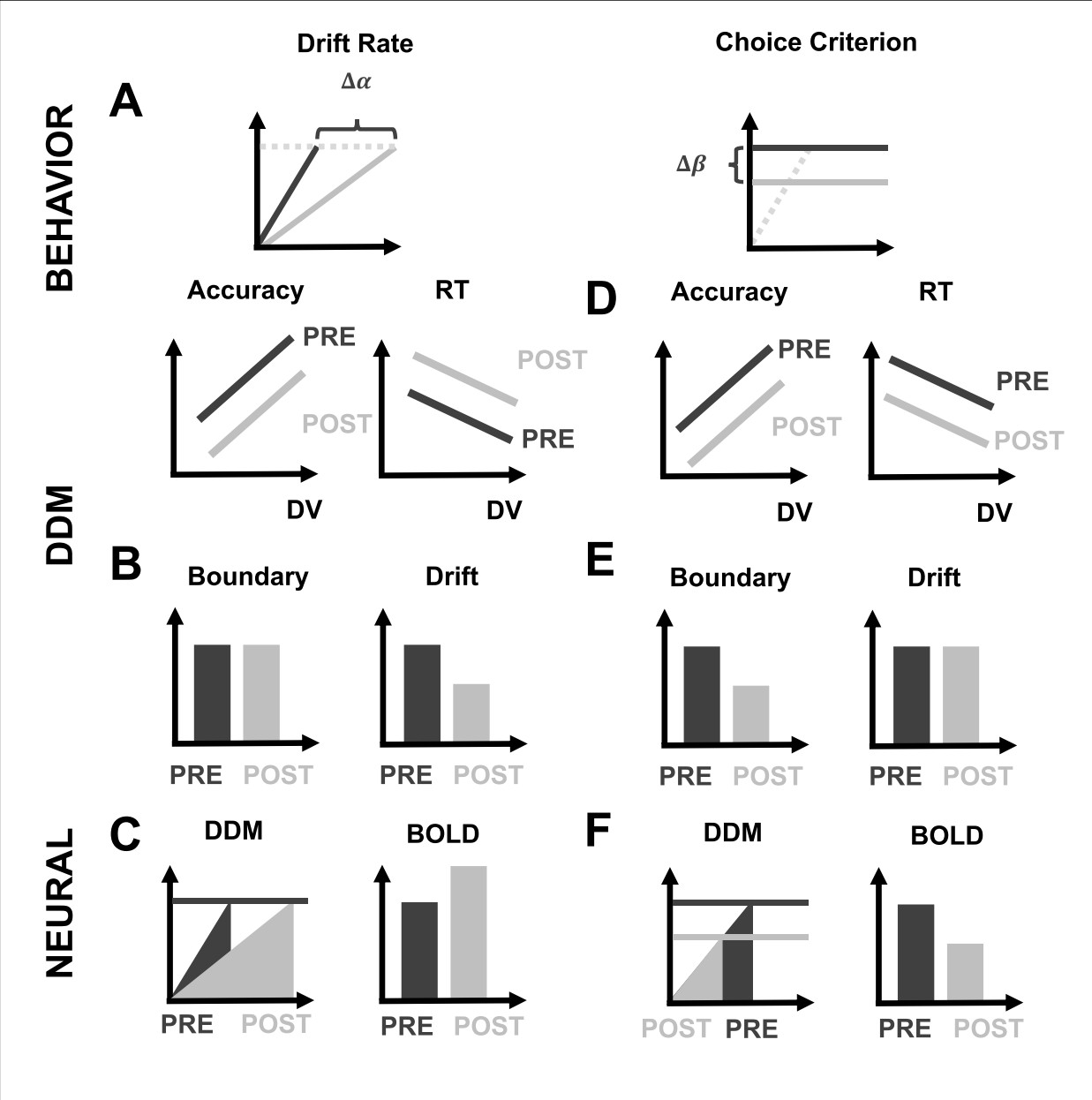

**Figure 2.** Study hypotheses. Scenario 1: left superior frontal sulcus (SFS) is causally involved in evidence accumulation. Theta-burst induced inhibition of left SFS should lead to reduced evidence accumulation (**A**), expressed as lower accuracy (A, second row, left), slowing of RTs (A, second row, right), and a reduction of DDM drift rate (B, right) without any effect on the boundary parameter (B, left). Since the neural activity devoted to evidence accumulation (area under the curve) should increase (C, left), we would expect higher BOLD signal in this case (C, right). Scenario 2: left SFS is causally involved in setting the choice criterion. Theta-burst induced inhibition of left SFS should lead to a lower choice criterion (**D**), expressed as lower choice accuracy (D, second row, left), faster RTs (D, second row, right), and a reduced DDM decision boundary parameter (**E**, left) without any effect on the DDM drift rate (E, right). At the neural level, we should observe reduced BOLD activity due to the lower amount of evidence processed by the neurons (F, right), and reflected by the smaller area under the evidence accumulation curve when it reaches the lower boundary (F, left).

The online version of this article includes the following figure supplement(s) for figure 2:

**Figure supplement 1.** Hierarchical Bayesian DDM.

**Figure supplement 2.** Neural-hierarchical drift-diffusion model (HDDM) alternatives.

choice to be taken. In this case, SFS inhibition should result in decreases in both choice precision and reaction times, a decrease in the DDM boundary parameter (*Rahnev et al., 2016*), and a reduction in associated neural activity due to the lower amount of evidence accumulated during the shorter response time (*Figure 2D–F*). Here, we directly test these two contrasting scenarios, by characterising the behavioural, neural, and neuro-computational consequences of cTBS to the left SFS. Crucially, we also investigate for both possible outcomes whether the functional contribution of the SFS during decision-making is indeed specific for perceptual choices, by comparing the results between the two matched types of choices.

## Behaviour: validity of task-relevant pre-requisites

Before scrutinising the role of the left SFS for either type of choice, we first behaviourally and neurally confirmed the validity of our task paradigm. To establish a fair comparison between perceptual (PDM) and value-based decision-making (VDM), we must necessarily show that we can distinctly identify the brain regions associated with each type of choice, and that behaviour is systematically a function of their respective evidence measures. Initial visual inspection shows that choice accuracy/consistency systematically increases (*Figure 2A*) and RTs become faster (*Figure 2B*) the larger the evidence difference, and this holds across tasks and stimulation conditions. All behavioural and fMRI analyses were performed on valid trials only (see Methods for inclusion criteria). Behavioural regressions confirmed that our task design allowed for a clear computational separation of both choice types: during perceptual decisions, participants relied exclusively on perceptual evidence, as reflected in both increased choice accuracy (main effect SD, $\beta = 0.560, p < 0.001$ and VD, $\beta = 0.023, p = 0.178$; *Figure 1B* and *Supplementary file 5*) and faster reaction times (RTs) with larger perceptual evidence, but not value-based evidence (main effect SD, $\beta = -0.057, p < 0.001$ and VD, $\beta = 0.002, p = 0.281$; *Figure 1C* and *Supplementary file 5*). Conversely, participants relied only on value evidence during VDM, as evident from both choice consistency (main effect VD, $\beta = 0.249, p < 0.001$ and SD, $\beta = 0.005, p = 0.826$; *Figure 1B* and *Supplementary file 5*) and RTs (main effect VD, $\beta = -0.016, p = 0.011$ and SD, $\beta = -0.003, p = 0.419$; *Figure 1C* and *Supplementary file 5*) irrespective of the items' SD. Thus, our results replicate previous findings obtained with a similar paradigm (*Polanía et al., 2014*; *Polanía et al., 2015*; *Grueschow et al., 2015*) showing that participants can use exclusively task-relevant evidence to make choices, and they confirm the suitability of our paradigm for directly comparing perceptual- and value-based decisions with matched stimuli and motor responses. Across sessions, RTs tended to shorten in both tasks. In line with the hierarchical drift-diffusion model (HDDM) results—selective boundary reductions for PDM and selective non-decision times (nDT) shortening for VDM—we interpret the VDM RT speed-ups as reflecting more efficient non-decision (sensorimotor) components rather than changes in evidence accumulation. To illustrate baseline trends, we provide session-wise RT trajectories (see *Appendix 1—figure 1* for RT-by-session). For completeness, group-mean accuracies by task are provided descriptively in *Figure 3A*; inferential tests focus on evidence-specific effects and TMS-induced changes within task.

## fMRI: VDM and PDM distinctly recruit brain processes, while recruiting similar visual and motor processes

In line with the behavioural results that participants depended on different evidence for the two types of choices, initial fMRI analysis revealed that neural activations strongly differed between choice types, despite the fact that participants saw the same images and gave the same motor responses.

We performed two levels of fMRI analyses. The first analysis examined average BOLD activity at the task level, contrasting perceptual versus value-based decisions, irrespective of whether the differences were driven by evidence accumulation or other cognitive processes. This initial analysis aimed to broadly identify brain areas recruited for each choice task and those common to both. First, we found visual and motor areas were jointly activated for both types of choices (p < 0.05, FWE-corrected with cluster-forming thresholds at $T(19) > 2.9$; *Figure 1—figure supplement 1A* and *Supplementary file 2*). Second, PDM led to stronger recruitment of the posterior parietal cortex, whereas VDM led to stronger activations of the medial prefrontal cortex (PFC) and posterior cingulate cortex (*Figure 1—figure supplement 1B* and *Supplementary file 2*), all in line with previous findings (*Grueschow et al., 2015*).

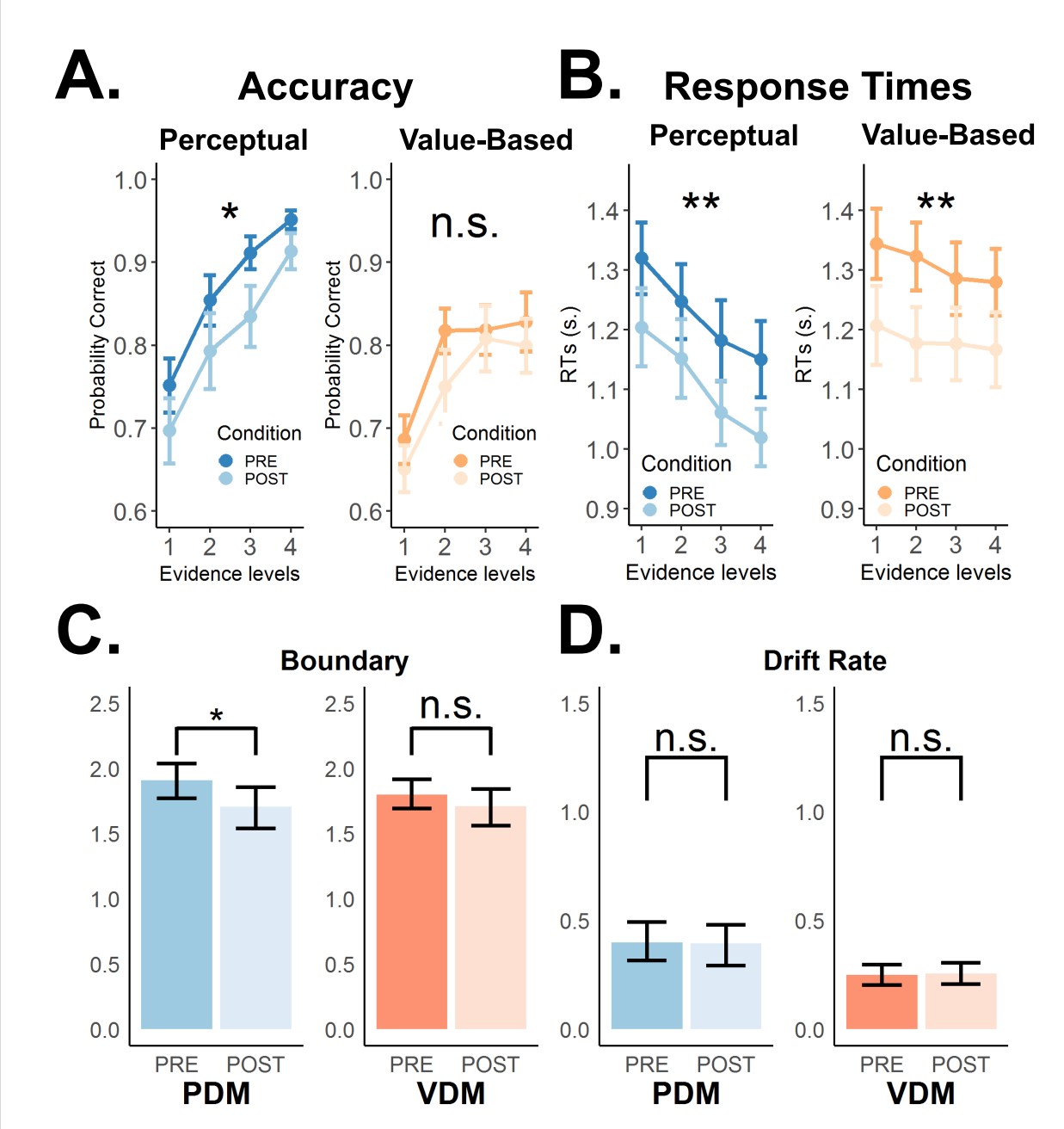

**Figure 3.** Theta-burst stimulation over the left superior frontal sulcus (SFS) affects choice behaviour and selectively lowers the decision boundary for perceptual but not value-based choices. (**A**) Choice accuracies/consistencies and (**B**) response times (RTs) for perceptual- (blue) and value-based (orange) decisions for different evidence levels during pre-cTBS (dark) and post-cTBS (light) stimulation periods. Error bars in (**A**) and (**B**) represent SEM. Consistent with previous findings, stronger evidence leads to more accurate choices and faster RTs in both types of decisions. Importantly, theta-burst stimulation significantly lowered choice accuracy selectively for perceptual, not value-based decisions (negative main stimulation effect for perceptual decisions and negative stimulation × task interaction; *Figure 3—figure supplement 1C* and see also *Figure 3—figure supplement 1A* for changes in choice accuracy across runs). Additionally, theta-burst stimulation also significantly lowered RTs in both choice types (negative main stimulation effect; *Figure 3—figure supplement 1C* and see also *Figure 3—figure supplement 1B* for changes in RTs across runs). (**C**) Theta-burst stimulation selectively decreased the decision boundary in perceptual decisions only (difference between estimated posterior population distributions; see Methods and *Figure 3—figure supplement 2A* for a detailed post hoc analysis). All the other parameters, particularly (**D**) the drift rate (see also *Figure 3—figure supplement 2B* for post hoc analysis), remain unaffected by stimulation. Error bars in (**C**) and (**D**) represent the 95% confidence interval range of the posterior estimates of the DDM parameters. $*p < 0.05$, $**p < 0.01$, and $***p < 0.001$.

The online version of this article includes the following figure supplement(s) for figure 3:

*Figure 3 continued on next page*

*Figure 3 continued*

**Figure supplement 1.** Theta-burst stimulation in left superior frontal sulcus (SFS) selectively lowers choice accuracy for perceptual decisions, but RTs become faster after stimulation in both choice types.

**Figure supplement 2.** Theta-burst stimulation in the left superior frontal sulcus (SFS) reduced decision boundary for perceptual decisions.

**Figure supplement 3.** The DDM disentangles the latent decision-relevant and decision-irrelevant processes observed with faster RTs.

**Figure supplement 4.** Simulations of fitted model: theta-burst stimulation in the left superior frontal sulcus (SFS) reduced decision times and accumulated evidence.

Notably, the left SFS does not yet appear in this contrast analysis. This is because this initial analysis did not include the parametric modulator for evidence accumulation. At this stage, we were evaluating whether the brain can flexibly recruit distinct brain regions based on task design, irrespective of whether these differences arise from variations in evidence accumulation or other cognitive processes. The second analysis assessed how BOLD activity is modulated by trial-by-trial evidence strength. Here, we identified a parametric modulator for evidence accumulation and regressed it on BOLD activity using a general linear model (GLM). This analysis revealed that the left SFS shows significant activation specifically when perceptual evidence is parametrically modulated. BOLD activity in the left SFS is detectable only when perceptual evidence is considered6. In fact, previous studies have similarly shown that the SFS only appears once a variable that measures the degree of evidence accumulation is included in the analysis6. Thus, these choice-type-specific brain activations, in response to identical visual input and motor output, ascertain that participants recruit task-specific brain regions depending on the choice domain.

## Behaviour: theta-burst stimulation reduces choice accuracy for perceptual decisions only

Our results support the hypothesis that the SFS has a specific role for perceptual decision-making, on several experimental levels. Using a differences-in-differences (DID) logistic regression (Methods), we found that SFS-cTBS led to a significant decrease from pre- to post-cTBS blocks in accuracy for PDM (main stimulation effect, $\beta = -0.465 \pm 0.342, p = 0.008$; *Figure 3A*, *Figure 3—figure supplement 1A*), while VDM choice consistency remained unaffected by SFS stimulation ($\beta = -0.042 \pm 0.205, p = 0.691$; *Figure 3A*, *Figure 3—figure supplement 1A*). These differences were significant in direct comparison (stimulation × task interaction, $\beta = -0.094 \pm 0.087, p = 0.034$; *Figure 3A*; *Figure 3—figure supplement 1C* and *Figure 3—figure supplement 4*) fatigue or habituation effects after checking that the average accuracies in PDM were actually recovering in the second post-stimulation session while there was no change in choice consistency at all during VDM (*Figure 3—figure supplement 1A*). Interestingly, our DID linear regression (Methods) revealed that SFS-cTBS had comparable effects on reaction times in both tasks: faster RTs were observed after SFS-cTBS for both PDM (main stimulation effect, $\beta = -0.116 \pm 0.067, p = 0.003$; *Figure 3B*, *Figure 3—figure supplement 1B*) and VDM (main stimulation effect, $\beta = -0.125 \pm 0.063, p = 0.001$; *Figure 3B*, *Figure 3—figure supplement 1B*), with no significant difference between these two effects (stimulation × task interaction, $\beta = 0.009 \pm 0.069, p = 0.795$; *Figure 3B*; *Figure 3—figure supplement 1C* and *Supplementary file 6*). Overall, the specific changes in choice accuracy indeed reflect cTBS disruption in left SFS in perceptual decisions. At the same time, the common changes in RTs from the first to the second half of the experiment may not necessarily reflect TMS-related changes in SFS function but rather general training effects common to both tasks (*Mawase et al., 2018*), but this possibility can only be examined in more detail with computational modelling.

## Modelling: SFS-TMS reduces decision boundary only for perceptual decisions

To examine in detail which specific latent decision process was affected by SFS-cTBS, we fit the HDDM simultaneously to the accuracy and RT data of our participants. This canonical model of choices allowed us to identify and disentangle the effect of stimulation on various latent variables representing distinct components of the choice mechanism (*Ratcliff and Smith, 2004*; *Ratcliff and McKoon, 2008*; *Polanía et al., 2015*; *Figure 2—figure supplement 1* and see Methods).

To investigate the underlying processes through which the cTBS stimulation induced the observed behavioural changes, we fitted a hierarchical Bayesian drift-diffusion model (HDDM; see Methods) simultaneously to the accuracy and RT data of our participants (*Figure 3B–D*). Critically, we used the DDM parameters to identify and disentangle the effect of stimulation on choice accuracy from that on RTs (*Ratcliff and Smith, 2004*; *Ratcliff and McKoon, 2008*; *Polanía et al., 2015*; *Figure 2— figure supplement 1* and see Methods). A specific focus of this analysis was on whether SFS-cTBS would change the way participants set the choice criterion (decision threshold; *Rahnev et al., 2016*; *Bogacz et al., 2010*; *Domenech and Dreher, 2010*; *Herz et al., 2016*) or the efficiency with which choice-relevant evidence is accumulated (drift rate, *Philiastides et al., 2011*; *Basten et al., 2010*) (see Methods for more details and *Figure 1B, E*). We found that theta-burst stimulation selectively reduced the decision boundary in PDM (see Methods; $p_{mcmc} = 0.003$; *Figure 3C*, *Figure 3—figure supplement 2A*), while leaving the decision-relevant parameters, including the drift rate, unchanged ($p_{mcmc} = 0.822$ for drift rate; *Figure 3D*, *Figure 3—figure supplement 2B, C*). For VDM, by contrast, no effect of cTBS was observed for either of the two decision-relevant parameters ($p_{mcmc} = 0.115$ for boundary and $p_{mcmc} = 0.758$ for drift rate; *Figure 3C, D*, *Figure 3—figure supplement 2A–C*), supporting the specificity of the SFS involvement in perceptual decisions. Full posterior summaries are provided in *Supplementary files 8–11*, and model adequacy is confirmed by posterior-predictive checks of accuracies and RT distributions (*Appendix 1—figures 1–6*). However, we found that nDT was selectively reduced in VDM. Overall, our findings indicate that the left SFS is causally involved in modulating the decision threshold. This conclusion was further corroborated by direct comparison of these effects, which showed that SFS-cTBS had a significantly stronger impact on the boundary parameter for PDM compared to VDM (stimulation × task interaction for the decision threshold, $p_{mcmc} = 0.045$; *Figure 3— figure supplement 2A*; there were no such differences for drift rate; $p_{mcmc} = 0.685$; *Figure 3—figure supplement 2B*).

## Modelling: faster RTs during value-based decisions is related to non-decision-related sensorimotor processes

To address the underlying latent process driving RT effects in both choices, we examined other DDM parameters and measurements. The DDM assumes that RTs can be disentangled into a non-decision-related (nDT) component as well as decision times (DT). The nDT is a DDM parameter that indexes constant latencies associated with sensory and motor preparation processes that are invariant across trials with different choice evidence (*Verdonck and Tuerlinckx, 2016*; *Starns and Ma, 2018*); in other words, this parameter forms no part of the evidence accumulation process (*Feltgen and Daunizeau, 2020*; *White et al., 2018*) and may therefore reflect task learning processes from movement repetition (*Mawase et al., 2018*). In contrast, DT are the component of RT where evidence accumulation actually takes place, and we can measure and derive DT using the evidence-dependent DDM parameters (see Methods for more details).

Our results showed that the faster RTs observed for value-based decisions after the stimulation indeed did not reflect evidence-dependent choice processes, but rather a change in non-decision-related sensorimotor processes (nDT) (see Methods; *Figure 2—figure supplement 1*): this parameter was decreased after stimulation for VDM ($p_{MCMC} = 0.062$) but not PDM ($p_{mcmc} = 0.707$) (*Figure 3—figure supplements 2C and 3B*), with a significant difference between these effects ($p_{mcmc} = 0.041$; *Figure 3—figure supplement 2*). In contrast, estimated DT was smaller after stimulation during PDM ($p_{mcmc} = 0.003$; *Figure 3—figure supplement 4A*, left), but not VDM ($p_{mcmc} = 0.100$; *Figure 3—figure supplement 4A*, right). Taken together, these results suggest that the simultaneous change in RT reveals completely different computational processes, whereby faster RTs during value-based choice are simply a by-product of task-related learning that may perhaps be unrelated to stimulation, while faster RTs during perceptual choice are actually related to decision-relevant, evidence-dependent latent choice processes. However, completely ascertaining whether such effect from stimulation is due to SFS inhibition, we need clear causal evidence of changes from neural processing. The pattern of nDT and decision-time changes is consistent with the posterior-predictive fits shown in *Appendix 1—figures 1–6*, with numerical posterior summaries in *Supplementary files 8–11*.

# fMRI: SFS activation changes for perceptual choices in line with model predictions

To investigate whether our behavioural and computational results directly relate to task-specific disruption of neural activity in left SFS, we investigated BOLD response changes in this brain area after stimulation. We exploited the fact that our fitted DDM and its latent parameters make clear predictions about how BOLD responses in this area should change if the stimulation affects the neural computations involved in setting the boundary for the necessary amount of evidence accumulation. Importantly, these predictions translate to clear parametric regressors that we can use for trialwise analysis of fMRI data (*Basten et al., 2010*; *Domenech et al., 2017*; *Liu and Pleskac, 2011*). More specifically, we expected that the BOLD signal level is proportional to the DDM's accumulated evidence (*aE*), defined as the area below the modelled evidence accumulation curve up until the accumulator reaches the decision boundary (*Liu and Pleskac, 2011*; *Domenech et al., 2017*; *Basten et al., 2010*). Using subject-wise DDM-latent parameters, the average area below the decision boundary for each evidence level can be computed as a function of each participant's decision boundary divided by the mean drift rate (see *Figure 1C, F* and Methods for more details). Using the more detailed trialwise measures, however, the same area can be computed as a function of each trial's RTs divided by the evidence level, since according to the DDM, the duration of response times is directly proportional to the decision boundary, and the evidence level is directly proportional to the slope of the drift rate (*Ratcliff and Rouder, 1998*; *Ratcliff and McKoon, 2008*; see Methods for more details). Exploiting these two known facts from the DDM thus allows us to extend our test of the stimulation effect from individual-specific latent parameters to trialwise regressors and behavioural measures. Higher SFS BOLD signals are associated with higher *aE* and vice versa (*Basten et al., 2010*; *Liu and Pleskac, 2011*; *Filimon et al., 2013*; *Tosoni et al., 2008*), implying that a TMS intervention lowering the decision boundary should lower aE and therefore BOLD signals. Crucially, these latent changes predicted by the DDM should also be reflected in the subject-level simulations of accumulated evidence constructed from the DDM parameters.

Thus, we first tested whether our neural hypotheses would already be evident in the simulated trialwise aE regressors. We used individual parameters identified by fitting our computational framework to simulate expected neural activity on a trial-wise basis across participants. To this end, we derived the predicted aE from the model parameters for each participant. A comparison across cTBS and task conditions confirmed the predicted cTBS-related decrease in accumulated perceptual evidence for PDM ($p_{mcmc} = 0.003$; *Figure 4A* and *Figure 3—figure supplement 4*), the corresponding null effect for VDM ($p_{mcmc} = 0.100$; *Figure 4B* and *Figure 3—figure supplement 4b*), and a significant difference for this effect between both choice types (one-sided $p_{mcmc} = 0.048$; *Figure 3—figure supplement 4B*).

In the next step, we used the trial-by-trial accumulated evidence as a regressor in the statistical analysis of the BOLD signals, allowing us to test whether the left SFS shows the predicted changes in neural response to varying levels of perceptual evidence. First, we tested whether our predictor of neural accumulated evidence was represented in BOLD signals of similar task-specific areas as reported previously for PDM in SFS (*Heekeren et al., 2004*; *Heekeren et al., 2006*) and for VDM in ventromedial prefrontal cortex (vmPFC) (*De Martino et al., 2013*; *Grueschow et al., 2015*). This was confirmed by the data: During PDM, trialwise *aE* correlated with BOLD activity in the left SFS (peak at = −21, = 26, = 37; $SV < 0.05$; *Figure 4—figure supplement 1B* and *Supplementary file 3*) whereas, critically, no significant BOLD activity in the left SFS was observed during VDM. During VDM, *aE* related to BOLD activity in the vmPFC (peak at = 3, = 38, = −17; $SV < 0.05$; *Figure 4—figure supplement 1*) and the nucleus accumbens (peak at $x = 9$, $y = 11$, $z = −11$; $<0.05$, FWE-corrected with cluster-forming thresholds at $T(19) > 2.9$; *Figure 4—figure supplement 1*). For both types of choices, domain-general representations of aE were also evident (see *Figure 4—figure supplement 1* and *Supplementary file 3*).

We then tested whether cTBS specifically reduced the neural representation of accumulated perceptual evidence in the left SFS for PDM, as predicted by the behavioural and modelling results. In line with these predictions, comparison of the post–pre trial-aE regressor showed a lower BOLD response in left SFS to the trialwise perceptual evidence during PDM ($SV < 0.05$; *Figure 4C*, green patch). This effect was significantly stronger than the corresponding effect on evidence representations in this area during VDM ($SV < 0.05$; *Figure 4C*, blue patch). No effect was found for VDM alone.

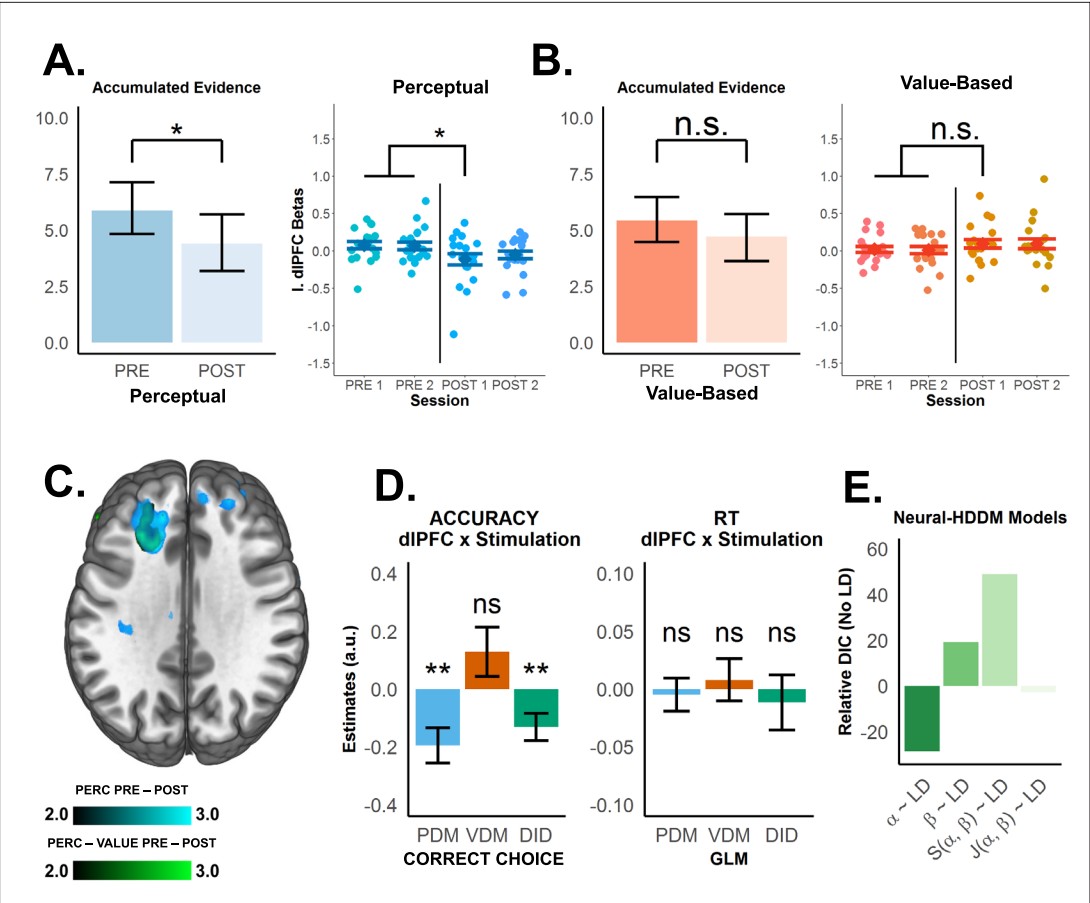

**Figure 4.** Neural representation of accumulated evidence in the left superior frontal sulcus (SFS) is disrupted after theta-burst stimulation and is linked with behaviour and neural computation. (**A**) Left panel: Accumulated-evidence (AE) simulation derived from the fitted DDM (left panel). Previous studies have illustrated how the accumulation-to-bound process convolved with the haemodynamic response function (HRF) results in BOLD signals; hence, the simulated AE provides a suitable prediction of BOLD responses in brain regions involved in evidence accumulation. Theta-burst stimulation selectively decreased AE for (**A**) perceptual (blue), not (**B**) value-based (orange) decisions (see *Figure 3—figure supplement 4B* for post hoc analysis). We constructed a trialwise measure of accumulated evidence using RTs and evidence strength for our parametric modulator (see Methods). Individual regions of interest (ROIs) extracted from the left SFS representing accumulated evidence across runs (right panels; see Methods) show that consistent with the DDM prediction, theta-burst stimulation selectively decreased BOLD response representing AE in left SFS during perceptual, not value-based decisions. Error bars in the left panels of (**A**) and (**B**) represent the 95% confidence interval range of the posterior estimates of the DDM parameters, while error bars in their respective right panels represent SEM. (**C**) Post–pre contrasts for the trialwise accumulated-evidence regressor show reduced left-SFS BOLD during perceptual decisions (green overlay), with a significantly stronger reduction for perceptual- versus value-based decisions (blue overlay). No reduction is observed for value-based decisions. (**D**) To test the link between neural and behavioural effects of transcranial magnetic stimulation (TMS), regression results show that after stimulation, BOLD changes in left SFS are associated with lower choice accuracy (left panel) for perceptual (PDM, blue) (negative left SFS × stimulation interaction) but not value-based choices (VDM, red), with significant differences between the effects on both choice types (difference-in-difference, DID, green, negative left SFS × stimulation × task interaction). On the other hand, cTBA-induced changes in left SFS activity are unrelated to changes in RT (right panel). Error bars in (**D**) represent the 95% confidence interval range of the estimated effect sizes. *$p < 0.05$, **$p < 0.01$, and ***$p < 0.001$. (**E**) To test the link between neural activity and DDM computations, we included trialwise beta estimates of left-SFS BOLD signals as inputs to the DDM. Alternative models tested whether trialwise left-SFS (LD) activity modulates the decision boundary ($\alpha$) (Model 1), the drift rate ($\beta$), or a combination of both (Models 3 and 4, see Methods and *Figure 2—figure supplement 2* for more details). Model comparisons using the deviance information criterion (DIC, smaller values mean better fits) showed that Model 1 fits the data best, confirming that the left SFS is involved in selectively changing the decision boundary for perceptual decisions.

The online version of this article includes the following figure supplement(s) for figure 4:

**Figure supplement 1.** Neural representations of accumulated evidence across the whole brain for PDM and VDM.

**Figure supplement 2.** Reanalysis of latent DDM parameters using the neural-hierarchical drift-diffusion model (HDDM) confirmed the results of lower decision boundary in perceptual decision-making (PDM) and lower non-decision times in value-based decision-making (VDM).

**Figure supplement 3.** Reanalysis of decision times and accumulated evidence using the neural-hierarchical drift-diffusion model (HDDM) provides improvements in model evidence and clearer statistical inference.

This indicates that the TMS effect is specific to the SFS during perceptual decisions, not value-based ones, as supported by the BOLD activity analysis. The reduction in BOLD activity in the left SFS during PDM after TMS is consistent with the DDM prediction of a reduction in the accumulated evidence due to a lower decision boundary. Convergent evidence for the specificity of this effect was provided by an alternative hypothesis-guided region-of-interest (ROI) analysis of the regression weights extracted from an a priori ROI mask of the SFS (see Methods). This showed lower post-stimulation beta values for the trial-*aE* regressor during PDM (main stimulation effect, $\beta = -0.153 \pm 0.054, p = 0.004$; *Figure 4A*) but not VDM (main stimulation effect, $\beta = 0.078 \pm 0.053, p = 0.140$; *Figure 4B*) and a significant difference in these effects (stimulation × task interaction, $\beta = -0.232 \pm 0.075, p = 0.002$; *Figure 4A, B*). Thus, the fMRI results show that cTBS of the left SFS indeed affects neural processing in this brain structure selectively during perceptual choices, in a way that is consistent with a lowering of the boundary and less accumulated evidence as predicted by the fitted DDM model. This remarkable convergence between the behavioural, modelling, and fMRI results suggests that the left SFS is indeed causally involved in setting decision criteria for choices based on perceptual evidence, but not based on subjective values.

## fMRI and modelling: neural-HDDM shows that perceptual-choice accuracy and boundary setting reflect trial-by-trial changes in SFS activity

If perceptual-decision performance depends specifically on activity in the left SFS, then trial-wise choice accuracy should relate to trial-wise BOLD activity in the SFS during perceptual decisions, over and above the mean effects of evidence level. To test this, we regressed choice accuracy/consistency on trial-by-trial BOLD activity extracted from the left SFS ROI, choice type, and TMS, while controlling for the evidence provided by the stimulus pairs on each trial (see Methods for details). In line with our prediction, we observed that the relation between perceptual-choice accuracy and trial-by-trial SFS activity was significantly decreased by TMS (SFS × stimulation interaction, $\beta = -0.196 \pm 0.128, p = 0.003$; *Figure 4D* and *Supplementary file 7*), independently of the corresponding effects for choice evidence (SD main effect, $\beta = 0.524 \pm 0.082, p < 0.001$, VD main effect, $\beta = 0.197 \pm 0.012, p = 0.001$, SFS × SD interaction, $\beta = -0.041 \pm 0.046, p = 0.365$, SFS × VD interaction, $\beta = 0.055 \pm 0.041, p = 0.183$). This effect was clearly specific for PDM, since no such effects were observed for VDM (SFS × stimulation interaction $\beta = 0.099 \pm 0.242, p = 0.422$; SFS × stimulation × task interaction, $\beta = -0.072 \pm 0.051, p = 0.005$; *Figure 4D* and *Supplementary file 7*) and for RTs during both types of choices (SFS × stimulation interaction, perceptual: $\beta = -0.031 \pm 0.053, p = 0.367$; *Figure 4D* and *Supplementary file 7*; accuracy: SFS × stimulation interaction, $\beta = -0.012 \pm 0.050, p = 0.650$; *Figure 4D* and *Supplementary file 7*).

We further investigated whether the relation between trialwise SFS activity and choice outcome indeed reflected an SFS role for perceptual boundary setting, as suggested by the DDM results presented above. To confirm this neurally, we set up several DDMs with trialwise SFS activity as an additional modulator for DDM parameters (on top of choice evidence; see Methods and *Herz et al., 2016*; *Herz et al., 2017*; *Turner et al., 2015*). More specifically, we tested several neural-DDMs in which trialwise SFS activity either modulated the decision threshold only (Model 1; *Figure 2—figure supplement 2A*), the drift rate only (Model 2; *Figure 2—figure supplement 2B*), or both parameters separately (Model 3; *Fiugre 2—figure supplement 2C*) or jointly (Model 4; *Figure 2—figure supplement 2D*). We compared these neural HDDMs to our baseline HDDM without neural inputs (see Methods for more details and *Figure 2—figure supplement 1*), allowing us to test across all conditions and choice types whether model evidence was enhanced when adding a potential trial-by-trial influence of SFS activity to the experimental inputs. Thus, the reported model evidence criterion (DIC) provides an additional formal test of whether the cTBS-influenced SFS activity relates selectively to the decrease of the decision boundary for perceptual choices only. Consistent with this prediction, Model 1, where SFS activity modulated the decision threshold only, outperformed all other models and model evidence showed improvements versus the baseline model (relative $DIC = -28.65$; *Figure 4B*). These results provide direct evidence that neural computations in the left SFS support criterion setting for perceptual evidence accumulation (see also *Figure 4—figure supplements 2 and 3*).

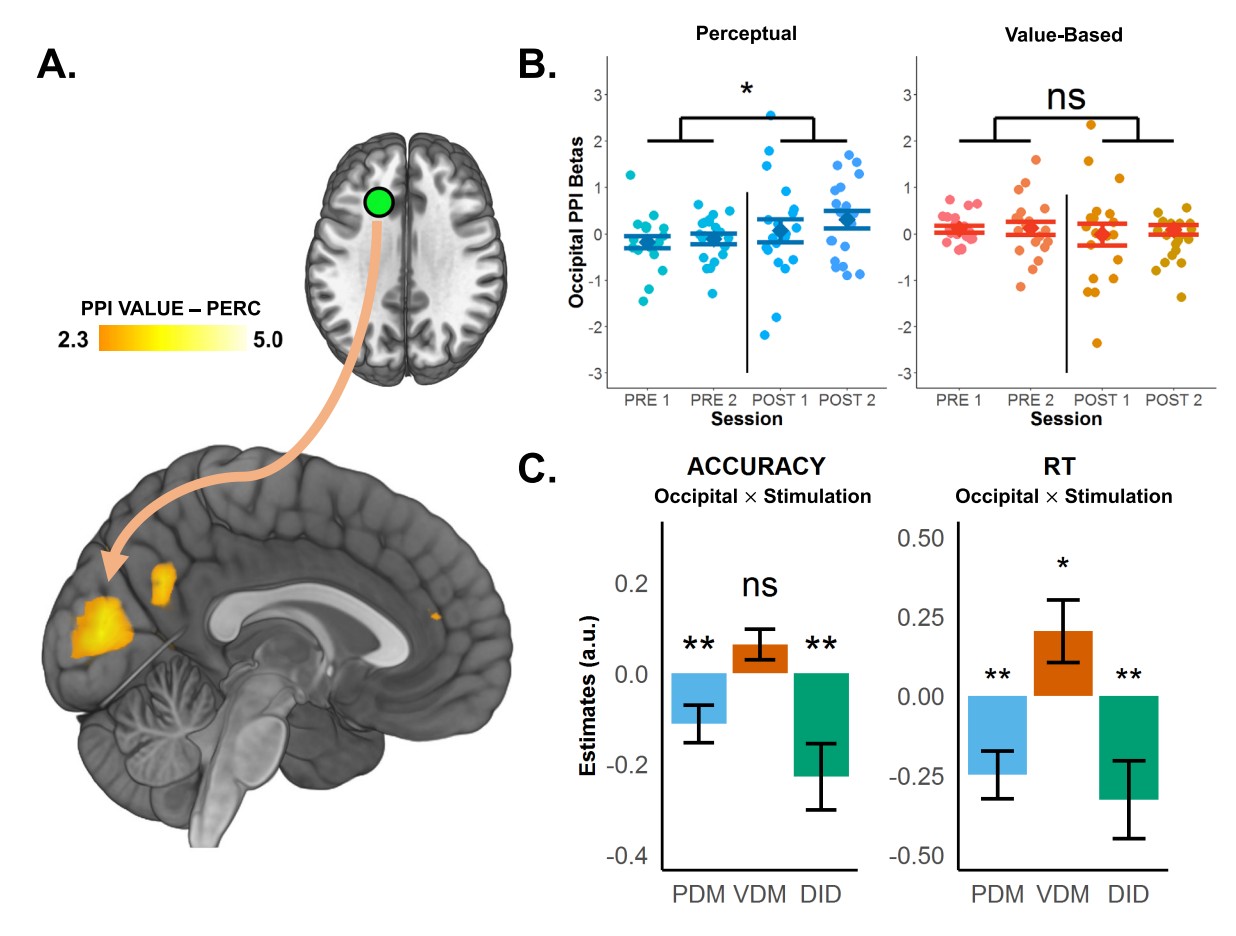

**Figure 5.** SFS-TMS (superior frontal sulcus-transcranial magnetic stimulation)-related changes in behaviour and neural computations are accompanied by increased functional coupling between the left SFS and occipital cortex (OCC). (**a**) Psychophysiological interaction (PPI) analysis reveals an area in OCC showing increased functional coupling with the left SFS during perceptual choices. (**b**) Region of interest (ROI) analysis of individual PPI betas shows that *aE*-related functional coupling between the left SFS and OCC is selectively increased post stimulation during perceptual (left panel) but not value-based decisions (right panel). Error bars in (**B**) represent SEM. (**C**) Regression results testing the link between continuous theta-burst stimulation (cTBS) effects on left SFS–OCC functional coupling and behaviour. Increased SFS–OCC coupling is associated with lower choice accuracy (left panel) specifically for perceptual (PDM, blue, negative OCC × stimulation interaction) but not value-based choices (VDM, red). In addition, increased functional coupling is also associated with faster RTs (right panel) for perceptual (blue, negative OCC × stimulation interaction) and slower RTs for value-based choice (red, positive OCC × stimulation interaction). Error bars in (**C**) represent the 95% confidence interval range of the estimated effect sizes. $^*p < 0.05$, $^{**}p < 0.01$, and $^{***}p < 0.001$.

## fMRI and connectivity: TMS affects SFS functional connectivity during perceptual choices

Overall, our results clearly indicate that cTBS to the left SFS disrupts selectively a neural process related to setting the criterion for perceptual evidence accumulation. At this point, we consider the possibility that cTBS may conceivably change the functional communication of the SFS with other brain areas involved in initial processing of the perceptual information necessary to make a choice. We explored this possibility by investigating whether cTBS affected functional coupling of the SFS. A psychophysiological interaction (PPI) analysis seeded in left SFS and modulated by *aE* indeed revealed stronger coupling with occipital cortex (OCC) after cTBS (peak at $x = -28, y = -85, z = -2$; $p < 0.01$ FWE-cluster-forming thresholds at $T(19) > 2.9$; **Figure 5A**). Interestingly, the activity peak in visual cortex showing evidence-dependent coupling with SFS overlaps with the spatiotopic neural representation of the stimulus items in the visual field during decision-making. We identified this overlap using a conjunction analysis of the PPI result and a contrast regressing BOLD signal on trial-by-trial stimulus onsets of both choice types (at familywise-error-corrected thresholds). Moreover, we used the latter

contrast to define fully independent ROIs in OCC processing the visual stimuli independent of task type and performed an ROI analysis on the individual SFS–OCC–PPI betas extracted for each participant. This confirmed that evidence-related functional coupling is increased by stimulation during PDM (main stimulation effect, $\beta = 0.330 \pm 0.284, p = 0.022$) but not VDM (main stimulation effect, $\beta = -0.186 \pm 0.247, p = 0.139$; *Figure 5B*; stimulation × task interaction, $\beta = 0.517 \pm 0.44, p = 0.021$; *Figure 5A*). Thus, our exploratory analysis shows that cTBS to the left SFS leads to stronger functional coupling with occipital areas involved in processing the visual stimuli, perhaps consistent with increased downstream demand on visual-related resources when upstream evidence accumulation regions are impaired.

We further explored whether this TMS-induced increase in functional coupling between the left SFS and OCC is related to changes in behaviour and specific neural computations during perceptual decisions. To test this, we related these effects to individual measures of choice behaviour and latent DDM parameters for each participant. This revealed that stimulation-induced increases in SFS–OCC coupling were associated with lower accuracy (OCC × stimulation × task interaction, $\beta = -0.225 \pm 0.142, p = 0.002$; *Figure 5C*) and shorter RTs (OCC × stimulation × task interaction, $\beta = -0.325 \pm 0.238, p = 0.007$; *Figure 5C*) for PDM, but not VDM. Taken together, these results show that the causal behavioural and computational changes during perceptual decision-making due to left SFS-TMS may relate not just to local neural changes in SFS, but also to the way this brain structure communicates with visual cortex.

## Discussion

Our study shows that the left SFS serves a domain-specific causal role in the accumulation of perceptual evidence, and particularly, the underlying computations affected by non-invasive brain stimulation is the setting of the choice criterion during perceptual, but not value-based choice. Our findings are more in line with that of *Rahnev et al., 2016*, who also suggested that SFS disruption leads to lower threshold setting. In contrast, previous work has also shown that SFS activity correlates with the evidence accumulation process reflected by the drift rate. These findings are not necessarily incompatible, but rather, they may reflect the interlinked computational mechanisms of the SFS in perceptual evidence accumulation, where the SFS plays a role in setting the decision threshold. Furthermore, our findings, in a way, contribute to closing the double dissociation gap left by previous work (*Polanía et al., 2015*), where it was shown that frontoparietal connectivity is causally specific to the precision of value-based, not perceptual choice using the same matched perceptual- and value-based choice task. More importantly, our study provided results above and beyond current standards of causal studies in humans (*Philiastides et al., 2011*; *Rahnev et al., 2016*; *Polanía et al., 2015*; *Murd et al., 2020*), that was only observed in animal studies (*Ding and Gold, 2012b*; *Erlich et al., 2015*; *Hanks et al., 2015*). Here, we simultaneously showed that causal TMS effects affected behaviour, latent computations and, more crucially, neural circuitry, as observed by changes in fMRI-BOLD activation after stimulation.

Many human decision neuroscience studies have employed model-based approaches to identify BOLD signals that correspond to computational processes (*Forstmann et al., 2011*; *Palmeri et al., 2017*; *Wijeakumar et al., 2017*). However, the links between neural and latent computational processing established by these studies are largely correlational (*Logothetis, 2008*; *Poldrack, 2006*; *Ramsey et al., 2010*), and there are many model alternatives that could possibly account for BOLD signals. Our study illustrates that causal manipulations induced by targeted functional inhibition of brain areas can provide decisive information and provide more direct support for neuro-computational mechanisms posited by cognitive models. Specifically, our study underlines that the DDM provides a plausible mechanistic account of the decision process (*Herz et al., 2016*; *Herz et al., 2017*; *Turner et al., 2015*), by showing that left SFS inhibition by cTBS affects the evidence representation posited by the model consistently across behavioural, computational, neural, neural–behavioural levels. Importantly, our results directly link changes in behaviour to changes in both latent computations and neural processing, by demonstrating how raw trialwise neural signals from the left SFS can augment the DDM to explain behaviour. This suggests that once brain stimulation studies have established (causal) correspondence between neural activity and latent variables in decision models, such models can be fruitfully extended by neural measures to provide a more complete characterisation and prediction of choice behaviour and potentially its malfunctions.

Our fMRI inferences rest on model-based assumptions linking accumulated evidence to BOLD amplitude. Alternative mechanisms—such as time-dependent (collapsing) boundaries—could attenuate the prediction that weaker-evidence trials yield longer accumulation and larger BOLD signals. While our behavioural and neural results converge under the DDM framework, we acknowledge this as a general limitation of model-based fMRI. The within-participant design enhances statistical sensitivity, yet the absence of an a priori power analysis constrains our ability to rule out small effects, particularly for null results in VDM. Consistent with this, model-fit indices and posterior-predictive checks indicated that the HDDM captured perceptual decisions somewhat more tightly than value-based decisions. This pattern likely reflects additional sources of variability in subjective value-based choices, together with our decision to keep the model architecture identical across tasks for principled comparison. Nevertheless, posterior-predictive simulations confirmed that, also for the value-based task, the model reproduced choice proportions and full RT distributions at the group and subject level (*Appendix 1—figures 1–6*), indicating that the fit quality is sufficient for the present inferences. Thus, although very small stimulation effects on value-based decision parameters cannot be completely excluded, the convergent behavioural, computational, and neural evidence points to a selective impact of cTBS on the perceptual decision process implemented in left SFS.

## Specificity of the SFS during perceptual decisions—only in humans?

To this end, our study reinforces already established findings by showing that the causal role of SFS during perceptual decisions (*Heekeren et al., 2004*; *Heekeren et al., 2006*; *Philiastides et al., 2011*; *Rahnev et al., 2016*) is a specialised function of evidence integration. Furthermore, our finding of a selective role of left SFS in perceptual evidence accumulation is particularly intriguing. The area appears to be uniquely developed in the human brain, with no close anatomical homologue in other species. In the animal literature, most prefrontal disruption studies in non-human primates have focused on the frontal eye fields (FEF) (*Ding and Gold, 2012a*; *Hanks and Summerfield, 2017*; *Shadlen and Newsome, 1996*) and in rodents on the frontal orienting fields (FOF) (*Erlich et al., 2015*; *Hanks et al., 2015*). While we and others observed disruption of the evidence accumulation process after interfering with SFS function in humans (*Philiastides et al., 2011*; *Rahnev et al., 2016*), disruption of the FOF in rodents has not affected behaviour at all or in a qualitatively different manner (*Brody and Hanks, 2016*; *Erlich et al., 2015*; *Hanks et al., 2015*). However, the results of electrical stimulation of the FEF in monkeys (*Ding and Gold, 2012a*; *Hanks and Summerfield, 2017*) cannot necessarily be directly compared with TMS studies of human SFS, since FEF and SFS in humans are both structurally and functionally distinct (*Murd et al., 2020*; *Rahnev et al., 2016*). Thus, while it is tempting to speculate that the SFS perceptual evidence accumulation process identified here may be specific to humans, it is possible that researchers may have to further consider other putative homologues across species that may truly correspond to the SFS area stimulated here (*Brunton et al., 2013*; *Hanks and Summerfield, 2017*).

## Do value-based decisions also rely on distinct PFC areas?

At the same time, our findings do not rule out the possibility of SFS involvement during value-based choice, where it may have a secondary function (but one that does not involve value evidence accumulation) or other specialised roles in decision-making. An alternative interpretation is that cTBS to left SFS induces a general speeding of perceptual processes that are engaged in both tasks. Under this view, the same perturbation of a perceptual decision stage could manifest as a reduction in decision boundary in the pure perceptual task, yet be absorbed as a shortening of nDT in the value task when modelled with a single accumulator. Our data cannot completely rule out this multi-stage account for the value task. Importantly, however, such a scenario remains compatible with our central conclusion that SFS plays a primarily perceptual role: any contribution of SFS to the value task would arise via an upstream perceptual component, while the value-comparison process proper is likely implemented in distinct valuation circuitry. The task-specific modulation of BOLD activity in SFS by cTBS during perceptual, but not value-based, choice—together with preserved value sensitivity and the absence of value-related SFS signals—strongly supports this interpretation.

Beyond our paradigm, previous work has suggested that during value-based decisions, the dlPFC more broadly interacts with the vmPFC to modulate value signals and facilitate self-control (*Hare et al., 2009*). Moreover, value-based decision-making encompasses a wide range of choice types

with varying degrees of complexity. For instance, more complex value-based decisions entail choices under risk (*Glickman et al., 2019*), intertemporal choice (*Peters and D'Esposito, 2020*), and strategic or social decisions (*Hutcherson et al., 2015*), which may plausibly recruit PFC regions due to working memory demands (*Barbey et al., 2013*), adjustment of decision time (*Sokol-Hessner et al., 2012*), or cost–benefit computations (*Basten et al., 2010*). In light of these additional value-based decision types, the functional specificity of SFS to perceptual decisions that we propose here should be viewed with some caution. Future studies should therefore compare more complex value-based decisions with perceptual decisions, while carefully matching task demands and complexity across domains to avoid confounds induced by context or difficulty.

## The role of SFS in choice criterion setting during perceptual decisions

That being said, the same holds true with the varying degrees of complexity across perceptual decision-making processes. Our findings show that the mechanism of which the left SFS is causally involved is modulating the decision threshold, with clearly consistent results across behavioural, computational, neural, and neural–behavioural levels. Our findings are more in line with that of *Rahnev et al., 2016*, who also suggested that SFS disruption leads to lower threshold setting. In contrast, previous work has also shown that SFS activity correlates with the evidence strength in the accumulation process, as reflected by the drift rate (*Basten et al., 2010*; *Heekeren et al., 2004*; *Heekeren et al., 2006*). In support of this notion, cortical activity disruption with repetitive transcranial magnetic stimulation resulted in lower choice accuracy and slower RTs (*Philiastides et al., 2011*). However, it is important to note that these results are not at all incompatible, but reflect the interlinked computational mechanisms of the SFS in perceptual evidence accumulation. All these studies, including ours, point out that the DDM's drift rate and decision boundary are both decision-relevant latent mechanisms with distinct as well as overlapping implications on choice behaviour: impairments in both boundary and drift lead to lower choice consistency (*Cavanagh et al., 2011*; *Green et al., 2012*; *Herz et al., 2016*; *Philiastides et al., 2011*; *Rahnev et al., 2016*). The main difference, in fact, concerns reaction times: lower drift rate implies slower RTs, while a lower boundary implies faster RTs. The differences in RT-TMS effects of our study and that of *Philiastides et al., 2011* may reveal the differences in task design and the nature of the stimuli. For instance, the study by *Philiastides et al., 2011* used dynamic series of briefly presented sequential face-house stimuli, varying the strength of each stimulus within noise to vary the evidence levels. By contrast, our study and that by *Rahnev et al., 2016* presented a static stimulus pair simultaneously during a 2AFC task and varied evidence not with noise but by stimulus difference.

We speculate that the goals of these tasks (i.e., discriminability from noisy dynamic stimuli versus stimulus SD between static stimuli) may sensitively affect different latent processes. Previous studies have shown that different tasks can produce proximally similar behaviour but may involve different goal functions and therefore, computationally distinct processes (*Heng et al., 2020*). For instance, noisy dynamic stimuli entail the accumulation of sensory evidence until the decision-maker can form representations suitable for choice discrimination *Ratcliff and McKoon, 2008*; thus, such stimuli are sensitively modulated by evidence strength and reflected by a lower drift rate when SFS is impaired (*Philiastides et al., 2011*) and where consequently, the circuitry takes more time to form perceptual representations. In contrast, evidence in the form of SD may depend on the sensitivity to which SFS can detect the difference, and this is modulated by the decision threshold (*Herz et al., 2016*; *Cavanagh et al., 2011*; *Green et al., 2012*; *Green et al., 2013*). Here, SFS impairment to accumulate evidence in the context of static, simultaneously presented stimuli may result in the inability of the circuitry to reliably discriminate SDs, resulting in the early termination of the accumulation process, which is behaviourally reflected as lower choice accuracy and faster RTs and computationally as lower thresholds. While a common evidence framework can plausibly reconcile our findings and that of previous studies, future studies should consider addressing this issue directly, by comparing the different goal functions and stimulus displays within perceptual decision-making tasks.

## Functional coupling between left SFS and visual cortex

Furthermore, our study further expands our understanding of SFS function vis-à-vis other brain regions. It is well established that the PFC is structurally connected with many other brain regions (*Wycoco et al., 2013*) and may flexibly interact functionally with different areas depending on choice

demands. Our exploratory connectivity results suggest that the SFS role for domain-specific accumulation of perceptual evidence is not just a local phenomenon but extends to functional communication with visual areas. Inhibition of this area's functional contribution to evidence accumulation led to an increase in its functional coupling with areas in OCC representing the stimuli visually upon which choices were based. The changes in functional coupling strength between the two cortical regions also corresponded to observed behavioural and latent computational changes. This suggests that perceptual choices rely not only on local processing in SFS but on an integrated functional circuit, comprising both SFS and OCC, at least for decisions based on visual stimuli as studied here. Though exploratory, our results are generally consistent with an occipito-frontal information exchange but extend it specifically to the SFS during perceptual evidence accumulation (*Bullier et al., 1996*).

We can speculate why the OCC may have been recruited after inhibition of the left SFS via cTBS stimulation. For example, it is possible that cTBS-related impairments in the accumulation mechanism implemented by the SFS bias the system to rely on second-best suboptimal mechanisms for solving the tasks, such as template matching from working memory. Previous work has provided converging evidence that maintenance of visual information in working memory enhances coupling between sensory processing in visual cortex and information storage in lateral PFC (*Gazzaley et al., 2007*; *Serences et al., 2009*). In fact, it has been suggested that SFS is canonically organised in 'memory receptive fields' (*Postle, 2016*) that may be more heavily taxed when direct accumulation mechanisms for sensory input are impaired, as in the case of cTBS manipulations. Of course, there are many other candidate mechanisms, such as attention or working memory, that may be more heavily taxed to compensate for the excitability manipulation of the SFS area specialised for processing the sensory evidence, as suggested by previous work on prefrontal–occipital interactions during various attention and working memory tasks (*Zanto et al., 2011*). Overall, the exploratory nature of our analysis warrants future investigation on the directionality of information flow between OCC and SFS. Additionally, future studies should also test whether perceptual choices based on other sensory modalities (e.g., touch, audition) lead to a flexible coupling of SFS with the specific sensory areas processing these stimuli. In any case, our study shows clearly that in the healthy, undisrupted human brain, left SFS plays a key role in transforming perceptual evidence into choices.

## Implications for theories of PFC organisation

Our study also contributes to our understanding of PFC functional organisation, given the considerable debates surrounding its organising principles (*Owen, 1997*). Previous studies have posited that different PFC regions contribute to specific aspects of information processing, in a manner that can be flexibly applied to all types of information, be it from different sensory modalities or in different cognitive formats (*Petrides, 2005*). Prevailing perspectives have also proposed an anterior-to-posterior hierarchy in PFC for the purpose of general cognitive control and executive function (*Nee and D'Esposito, 2016*), suggesting that the main role of the PFC is largely in the domain of higher-order cognitive and abstract operations that transcend specific functional domains (*Domenech and Koechlin, 2015*; *Koechlin and Summerfield, 2007*). In contrast, our finding of a domain-specific causal role of SFS in evidence accumulation for perceptual decision-making suggests that the PFC is organised as a collection of fractionated sub-regions, such that each region processes different types of information (*Goldman-Rakic and Leung, 2002*), as opposed to a systematic hierarchy. Moreover, the fact that the SFS is even involved in the integration of low-level perceptual evidence (*Heekeren et al., 2004*; *Heekeren et al., 2006*; *Heekeren et al., 2008*; *Philiastides et al., 2011*; *Rahnev et al., 2016*) implies that the PFC's role is not limited to higher-order cognitive function. Overall, our findings are in no position to propose an overarching framework of overall PFC organisation, given the limited area in posterior dlPFC targeted by our study, but rather a call to consider alternative views where the underlying organisational principles are more fractionated and less hierarchical.

## Implications for computational psychiatry

Finally, our finding of SFS causal involvement in decision threshold setting during perceptual decision-making may offer clinical implications. Particularly, manifestations of impulsive behaviour (*Burnett Heyes et al., 2012*) are largely apparent in clinical populations with aberrations in decision threshold setting (*Herz et al., 2016*). However, most studies of these disorders have focused on impulsive behaviour induced by reward or preferences (*Glimcher et al., 2007*). It is important to note here

that reward impulsivity is only one of the many domains of aberrant behaviour in clinical populations. Perceptual impulsivity is also important, since many of the behavioural and cognitive deficits are closely linked to impairments in perceptual function (*Fuermaier et al., 2018*). For instance, impulsive behaviour can also arise in non-reward-related settings, such as when perceptually discriminating SDs where less accurate and faster responses have been observed in people with addiction disorders (*Banca et al., 2016*) and borderline personality disorder (*Berlin and Rolls, 2004*). Perceptual deficiencies are also prevalent in clinical populations with attention-deficit hyperactivity disorder or Parkinson's disease and are thought to be linked to impairments in the dopaminergic system (*Fuermaier et al., 2018*). Prior causal evidence from deep-brain stimulation, in particular, has shown that disrupting the STN lowered decision thresholds, thus increasing this perceptual impulsivity among Parkinson's patients (*Herz et al., 2016*; *Herz et al., 2017*). Our findings that TMS of the left SFS causally and selectively lowered the decision boundary during perceptual decisions suggest that the lateral PFC may be functionally integrated with these cortico-striatal and cortico-subthalamic nuclei (STN) pathways (*Bogacz et al., 2010*; *Green et al., 2012*; *Green et al., 2013*).

Overall, impulsive behaviour is not exclusive to the reward domain, and our results suggest that there is something to gain from understanding impulsive behaviour in non-reward settings requiring decisions on perceptual information. Maladaptive behaviour may not only reflect individual wants or likings, often assumed by addiction studies, but could also be a function of low-level sensory or higher-order cognitive processes that have so far not fully been accounted for (*Fuermaier et al., 2018*). This may have serious implications for how cognitive therapies or interventions are designed, and our findings may provide useful insights in guiding such future work. Particularly, it is worth exploring to what degree the left SFS and its connections are structurally or functionally different in clinical populations, and whether these impulsive tendencies can be captured by sequential sampling models, such as the DDM.

# Materials and methods

## Participants

Twenty healthy right-handed volunteers (ages 20–30; 8 female) with normal or corrected-to-normal vision participated in the study. Participants completed the MRI screening and consent forms upon their arrival. They then went into the behavioural testing room and read the instructions of the experimental tasks. Participants were fully informed about the study's objectives, the equipment used during the experiment, and MRI safety. We also obtained their consent on the data recorded from them. No participant suffered from any neurological or psychological disorder or took medication that interfered with their participation in the study. Participants received monetary compensation for participation and performance of the perceptual choices, as well as one food item to consume after the experiment depending on a random value-based choice trial. The experiments conformed to the Declaration of Helsinki and the experimental protocol was approved by the Ethics Committee of the Canton of Zurich.

## Experimental paradigm

We asked participants to refrain from eating for 3 hr before the start of the experiment. Our experiments took place between 0800 and 1900 hr during the day. The experiment consisted of two steps: (1) a rating task outside the scanner and (2) a decision-making task inside the scanner. During the rating task, we asked participants to provide perceptual- and value-based ratings of the same set of 61 food images using an on-screen slider scale. All of the food items were in stock in our lab and participants were informed about this via visual inspection. For perceptual ratings, participants rated—on a scale from 5% to 100% in steps of 5%—how much of the black background within the white square perimeter was occupied by the food item. For value-based ratings, participants rated—on a scale from 5 to 100 in steps of 5—how much they wanted to eat the presented food item at the end of the experiment. We instructed participants that the midpoint of the scale in value-based ratings indicated indifference.

After rating the food items, an algorithm selected a balanced set of perceptual- and value-based trials divided into four evidence levels, $E$. The evidence levels are based on the absolute difference between the average ratings of the food items paired in each trial. We define perceptual evidence as

the absolute SD between the two food items. On the other hand, we define value-based evidence as the absolute VD between the two food items. In particular, the evidence levels for perceptual trials, $E_p$, are:

$$E_p = \left| r_{\text{biggest}} - r_{\text{smallest}} \right| \in \{5\%, 10\%, 15\%, 20\%\},$$

while the evidence levels for value-based trials, $E_v$, are:

$$E_v = \left| r_{\text{best}} - r_{\text{worst}} \right| \in \{1, 2, 3, 4\},$$

where $r = \frac{r_1 + r_2}{2}$ is the average food item from the two ratings while $r_{\text{biggest}} - r_{\text{smallest}}$ and $r_{\text{best}} - r_{\text{worst}}$ represent the ratings' difference for the pairs presented for perceptual- and value-based choices, respectively.

Inside the scanner, participants performed the decision-making task for which they chose between two food items, based on whether they were accumulating perceptual or value-based evidence. We matched the visual sensory stimuli of the food items as well as their motor outputs across the two choice types. The only difference was the type of evidence participants had to accumulate to make a choice. Each trial started with presentation of a central fixation marker (length ~0.8°, height ~0.3°). Next, a centrally presented word indicated whether participants would perform a perceptual (word 'AREA') or value-based (word 'LIKE') choice. On the subsequent screen, the task cue was replaced by either the letter 'A' or 'L' (~0.2°) to remind participants that they were in a perceptual or value-based block, respectively. Two food items were simultaneously displayed, one above and one below the screen (y eccentricity 3.6°; a white square of 6° width surrounded each food item). Blocks alternated between perceptual- and value-based choices in a given session (7–9 trials per task-block). Participants pressed one of two buttons on a keypad with their right middle finger (upper item) or right index finger (lower item) to indicate their choice. On a given trial, participants had 3 s for their choice; otherwise, the trial would be regarded as a 'missed trial' and would not enter the analysis. Analyses were conducted on valid trials only, defined as trials with a registered response within the task's response window and passing pre-specified validity checks; trials without a response were excluded and not analysed. Participants made correct or consistent choices when they chose the food item with the higher rating as indicated in the double ratings task prior to entering the scanner. After the experiment, participants stayed in the room with the experimenter while they ate the food that was selected based on the participants' choice in one randomly selected VDM trial. During perceptual decision-making blocks, participants were rewarded with 0.5 CHF every time they correctly discriminated between the SD of the two food items presented on the monitor screen.

The experiment had a total of 256 trials divided into 8 sessions of 32 trials each. The first four sessions were pre-stimulation sessions where participants performed the task without stimulation. The last four sessions were post-stimulation sessions during which participants performed the choices with decreased neural excitability in the SFS due to the preceding cTBS. The 256 trials were fully balanced across all factors (trial type: perceptual or value-based; evidence levels: 1–4; correct response: up or down).

## Stimulation protocol

We applied cTBS (*Huang et al., 2005*; *Di Lazzaro et al., 2005*; *Di Lazzaro et al., 2008*) to exogenously induce cortical inhibition of our ROI, an area in the left SFS (MNI coordinates: $x = -24, y = 24, z = 36$) (*Heekeren et al., 2004*; *Philiastides et al., 2011*). Before the main fMRI experiment, we identified the stimulation site over the left SFS (MNI coordinates: $x = -24, y = 24, z = 36$) (*Heekeren et al., 2004*; *Philiastides et al., 2011*) based on previous studies and each individual's stimulation intensity. In an initial fMRI session, we acquired high-resolution T1-weighted 3D fast-field echo anatomical scans used for subsequent neuro-navigation (181 sagittal slices, matrix size = 256 × 256, voxel size =1 mm³, TR/TE/TI = 8.3/2.26/181 ms, 3T Philips Achieva). The hand area of the left M1 (motor hotspot) was determined by identifying the first dorsal interosseous (FDI) movement-evoked potentials (MEPs) induced by TMS pulses. We delivered single monophasic TMS pulses using a figure-of-eight coil attached to the TMS stimulator. We then marked an equidistant circular grid on each individual's anatomical MRI scan using a neuro-navigation system over the hand motor region, located at the anterior portion of the central sulcus. We localised the optimal motor hotspot as the point in the grid that elicited the

strongest FDI MEPs from TMS pulses. Once we selected the motor hotspot, we asked participants to activate their FDI by pressing their thumb and index finger at about 20% maximum force in order to obtain their active motor threshold (AMT). We defined the AMT as the minimal TMS intensity required to produce MEPs of ≥200 mV amplitude (measured with Magventure MRi-B91) in ≥5–10 consecutive pulses. We retested the AMT by visually inspecting the FDI twitches triggered by TMS pulses over the marked optimal hotspot. The average AMT outside the scanner was 52.35 ± 6.27% while the AMT inside the scanner was 52.91 ± 6.18%. We applied cTBS at an intensity of 80% of the individual's AMT. The cTBS protocol contained bursts of three pulses at 50 Hz. This protocol has been shown to reduce cortical excitability for at least 30 min (*Huang et al., 2005*). Every burst was repeated at a rate of 5 Hz, resulting in 200 bursts with a total of 600 pulses delivered within 40 s.

Before moving our participant into the scanner, we marked the motor hotspot as well as the stimulation site on a swimming cap fixed in position by straps. Participants wore this cap while they were inside the scanner. Before the start of the fifth session, participants received cTBS over the left SFS. We used a figure-of-eight MR-compatible TMS coil (MRi-B91) attached to a TMS stimulator. Participants returned to the scanner after receiving stimulation and proceeded to complete the last four sessions. On average, the post-TMS fMRI task started 228 ± 41 s after the end of theta-burst stimulation following established protocols from previous studies (*Knecht et al., 2003*; *Philiastides et al., 2011*; *Thut and Pascual-Leone, 2010*). Given the established timeline of cTBS effects (*Huang et al., 2005*), we expected the stimulation effects to weaken over time due to neural recovery. In line with established procedures, we treated the first two post-stimulation sessions as the actual post-cTBS period and the last two post-stimulation sessions as a recovery period (*Philiastides et al., 2011*).

## Differences-in-differences

We implemented a DID regression analysis to identify the causal relationships based on stimulation-induced neural inhibition in SFS. We used the identical DID for behavioural, computational, neuroimaging, and connectivity analyses. Here, we use the following notation: task conditions $Task$ (perceptual, $Task = 1$; value-based, $Task = 0$); stimulation conditions $TMS$ (pre- $TMS = 0$ and post- $TMS = 1$); $V$ is our variable of interest, which may be behavioural or neural; and, the causal treatment effect, $\phi\left(V \mid Task, TMS\right)$, takes the following form:

$$
\begin{aligned}
&\phi\left(V \mid Task, TMS\right) \\
&= \left[\mathbb{E}\left(V \mid TMS = 1, Task = 1\right) - \mathbb{E}\left(V \mid TMS = 0, Task = 1\right)\right] \\
&\quad - \left[\mathbb{E}\left(V \mid TMS = 1, Task = 0\right) - \mathbb{E}\left(V \mid TMS = 0, Task = 0\right)\right],
\end{aligned}
$$

where $\mathbb{E}\left(V \mid Task, TMS\right)$ is the expected value of the variable of interest, $V$, given task and TMS condition. The first difference on the right-hand side captures the average stimulation effect for PDM while the second difference captures the average stimulation effect for VDM. The overall difference assumes that if behaviour will be the same after stimulation, then there is no effect, $\phi = 0$ (*Angrist and Pischke, 2009*; *Bertrand et al., 2004*). But if there is a stimulation effect and it impairs behaviour or neural activity, then $\phi < 0$.

## Behavioural analyses for choice

We analysed the influence of cTBS on choice using a logit regression on choices, $\rho$ (correct = 1, incorrect = 0) over various regressors of interest, such as TMS condition, $TMS$ (pre-cTBS = 0, post-cTBS = 1); task, $Task$ (perceptual = 1, value-based = 0); its interaction ($Task \times TMS$), which measures the causal stimulation effect, $\phi$; and, other regressors, $X^k$ that we use as controls. This includes task-relevant evidence (SD for perceptual and VD for value-based, 1–4), response times (RTs), and task-irrelevant evidence (i.e., VD for perceptual and SD for value-based, 1–4). The full model is,

$$
\Pr\left(\rho_{t,c,s,i}^{DID}\right) = \frac{1}{1 + \exp\left(-\beta_0 + \beta_1 Task_{(t,c,s,i)} + \beta_2 Task_{(t,c,s,i)} + \phi Task_{(t,c,s,i)} TMS_{(t,c,s,i)} + \sum_{k=4}^{n} \beta_k X_{(t,c,s,i)}^k\right)},
$$

where $t$ indexes task, $c$ for TMS, $s$ for subject, and $i$ for trial. Since our model contains a DID interaction term, nonlinearity of the logit regression results is a non-zero estimate even if the true causal effect is zero, $\phi = 0$.

To remove nonlinearity bias and isolate the true causal effect, we ran another logit regression without the interaction term,

$$\Pr\left(\rho_{t,c,s,i}^{NODID}\right) = \frac{1}{1 + \exp\left(-\left[\beta_0 + \beta_1 Task_{(t,c,s,i)} + \beta_2 TMS_{(t,c,s,i)} + \sum_{k=4}^{n} \beta_k X_{(t,c,s,i)}^k\right]\right)},$$

and we take the difference between the two logits (**Ai and Norton, 2003**; **Karaca-Mandic et al., 2012**; **Puhani, 2012**),

$$\Pr\left(\rho_{(t,c,s,i)}^{TRUEDID}\right) = \Pr\left(\rho_{(t,c,s,i)}^{DID}\right) - \Pr\left(\rho_{(t,c,s,i)}^{NODID}\right).$$

We also ran variations of the model to test for robustness, particularly GLMs with or without control variables, and we also tested robustness using various stimulation runs (see **Supplementary file 5**). We used cluster-robust standard errors at the subject level under the assumption that each individual performance is independent across participants. We implemented this analysis using STATA/SE 13.1.

## Behavioural analyses for response times

We similarly used DID regressions to analyse the influence of cTBS on response times (*rt*). Here, we simply ran a GLM for our regression,

$$rt_{t,c,s,i} = \beta_0 + \beta_1 Task_{(t,c,s,i)} + \beta_2 TMS_{(t,c,s,i)} + \phi Task_{(t,c,s,i)} TMS_{(t,c,s,i)} + \sum_{k=4}^{n} \beta_k X_{(t,c,s,i)}^k + \varepsilon_{(t,c,s,i)}$$

and we also ran variations of the model (see also **Supplementary file 5**). We similarly used cluster-robust standard errors at the subject level.

## Hierarchical Bayesian DDM

We analysed the effect of cTBS on PDM and VDM using HDDM. The model assumes evidence is accumulated through a one-dimensional Wiener process, whereby the state of evidence, $x_t$ at time $t$ evolves through a stochastic differential equation,

$$\frac{dx_t}{dt} \sim \mathbb{N}\left(\delta, \sigma^2\right).$$

Here, $\delta$ is the amount of evidence being accumulated at time $t$,

$$\delta_{c,s,i} = \kappa_{c,s} \times E_{c,s,i},$$

where $E$ represents the evidence level and $\kappa$ is the drift rate that linearly scales the evidence and this is typically interpreted as quality of information processing. Thus, $\kappa_{c,s}$ is the only free drift parameter per subject and condition; any variation in $\delta$ across evidence levels is determined by the corresponding evidence values $E_{c,s,i}$ rather than by separate drift parameters for each level. When we report drift values by evidence level (**Supplementary files 8 and 9**), these entries summarise the posterior mean of $\delta_{c,s,i}$ for trials in each evidence bin implied by this $\kappa \times E$ formulation. Furthermore, the model assumes evidence is accumulated at the starting point, $\beta$, and the accumulation process continues until a choice, $\rho$, is made at time $t_d$ at a given threshold, $\alpha$. Once the accumulation process terminates, the state of evidence is either $x_t > \alpha$ (a correct decision) or $x_t \leq 0$ (an incorrect decision). We also accounted for visual sensory processing and motor response delays with the non-decision time parameter (*nDT*), $\tau$.

The hierarchical Bayesian model is implemented whereby each observed choice, $y_{c,s,i}$ $(\rho, rt)$, follows a Wiener distribution, $\omega$,

$$y_{(c,s,i)} \sim \omega\left(\delta, \alpha, \tau, \beta\right),$$

where $c$ indexes task ($c = p$ for perceptual, $c = v$ for value-based), $s$ for participants ($s = 1, \ldots, N_{subjects}$), and $i$ for trials ($i = 1, \ldots, N_{trials}$). Furthermore, the hierarchical structure contains three random variations at the trial, subject, and condition levels. We treated all interindividual differences per stimulation condition level as random effects:

$$\delta_{(c,s,i)} \sim N\left(\mu_{\delta(s)} E_{(c,s,i)}, \sigma^2_{\delta(s)}\right),$$

$$\tau_{(c,s,i)} \sim N\left(\mu_{\tau(s)}, \sigma^2_{\tau(s)}\right),$$

$$\alpha_{(c,s,i)} \sim N\left(\mu_{\alpha(s)}, \sigma^2_{\alpha(s)}\right),$$

where $N(\mu, \sigma)$ is a normal distribution with mean, $\mu$ and standard deviation, $\sigma$. Here, $E$ represents the trial-by-trial evidence levels, which we measure in absolute terms; and we fix the starting point, $\beta_{c,s,i} = 0.5$. We used Bayesian hypothesis testing to compare posterior probability densities.

## Measure of accumulated evidence

We computed estimates for DT ($t_{d(c,s)}$) and accumulated evidence ($aE_{c,s}$) to test whether $aE$ is a plausible representation of the accumulation process at the neural level. Following the literature (**Bogacz et al., 2006**; **Bogacz et al., 2010**), we define mean decision time as the ratio between the decision threshold and the drift rate shaped by a hyperbolic tangent function,

$$t_{d(c,s)} = \left(\frac{\alpha_{c,s}}{\kappa_{c,s}}\right) \tanh\left(\kappa_{c,s} \times \alpha_{c,s}\right).$$

It is important to note that reaction time, $rt$, is the sum of both DT and nDT, $rt = t_d + \tau$.

We define accumulated evidence ($aE$) as the area below the drift process up until the accumulator reaches the decision boundary:

$$aE_{c,s} = \frac{\alpha_{c,s} \times t_{d(c,s)}}{2}.$$

Here, we derive $aE$ using the area equation of a triangle, where decision time $t_{d(c,s)}$ is the base and the decision boundary, $\alpha_{c,s}$, is the height.

## MCMC sampling

To estimate all parameters, we performed Gibbs sampling via Markov Chain Monte Carlo (MCMC) in JAGS (**Plummer, 2016**) to generate parameter posterior inferences. We drew a total of 100,000 samples from an initial burn-in step and subsequently drew a total of new 100,000 samples with three chains each. We derived each chain based on different random number generator engines with different seeds. We applied a thinning of 100 to this final sample, resulting in a final set of 1000 samples for each parameter. This thinning assured auto-decorrelation for all latent variables of interest. We conducted Gelman–Rubin tests for each parameter to confirm chain convergence. All latent parameters in our Bayesian model had $\hat{R} < 1.05$, suggesting that all three chains converged to a target posterior distribution. We compared the difference in posterior population distributions estimated for each parameter between the stimulation conditions as well as the DID, which included differences between tasks. We tested whether the resulting distribution (i.e., the causal stimulation effect) is significantly different from zero (i.e., the null hypothesis) using the cumulative function up to or from 0 depending on the direction of the effect. We refer to this probability as Bayesian 'p-values', $p_{mcmc}$.

## fMRI data analysis

Participants performed eight choice-task sessions while BOLD images were recorded with a Philips Achieva 3T whole-body scanner. We used statistical parametric mapping (SPM8, Wellcome Trust Centre for Neuroimaging) for image pre-processing and analysis. In particular, images were slice-time corrected (to the acquisition time of the middle slice) and realigned to account for subjects' head motion. Each participant's T1-weighted structural image was co-registered with the mean functional image and normalised to the standard T1 MNI template using the new-segment procedure in SPM8. The functional images were normalised to the standard MNI template using the same transformation, spatially resampled to 3 mm isotropic voxels, and smoothed using a Gaussian kernel (FWHM, 8 mm).

We estimated two GLMs, constructed by convolving a series of appropriately placed indicator functions with the default model of the BOLD response embedded in SPM8. GLM1 contained only two indicator functions for the onsets of PDM or VDM trials. On the other hand, GLM2 contained four

indicator functions for the onsets of task (PDM and VDM trials) and stimulation (pre- or post-TMS) runs, coupled with one regressor each for parametric modulation of the BOLD response by the trial-wise accumulated evidence (aE). We earlier demonstrated that the theoretical average accumulated evidence is derived from population-level as well as subject-level latent DDM parameters, by dividing the estimated decision boundary by the estimated drift rate. To construct a trialwise measure of $aE$, we exploit the fact that the length of the RTs is directly proportional to the size of the decision boundary while the evidence level, $E$, is directly proportional to the drift rate (**Basten et al., 2010**; **Domenech et al., 2017**; **Kiani et al., 2014**). With this mapping, we can then construct a parametric trialwise measure of accumulated evidence, $aE_{t,c,s,i}$,

$$aE_{t,c,s,i} = \sqrt{\frac{RT_{t,c,s,i}}{E_{t,c,s,i}}},$$

where the square root function accounts for the concave nonlinearity in accumulated evidence. Previous work (**Tajima et al., 2016**) has shown theoretically that the shape of the accumulated evidence is indeed concave, where it suggests that the rate of accumulating evidence is decreasing as the decision process continues to accumulate. This concavity in $aE$ is consistent with DDM predictions where evidence accumulation is steeper during earlier responses and begins to plateau at later responses (**Ratcliff and McKoon, 2008**; **Ratcliff and Smith, 2004**).

We convolved our GLMs with a canonical haemodynamic response function, modelled MR image autocorrelations with first-order autoregressive model, and included 6 motion parameters (obtained during realignment) as regressors of no interest. After fitting the model to the BOLD data, we tested regressors for statistical significance at the second level, in random-effects group one-sample $t$-tests of the corresponding single-subject contrast images. We performed statistical inference at the cluster level, using whole-brain family-wise-error-corrected (FWE-corrected) statistical threshold of $p < 0.05$, based on a cluster-forming voxel cutoff at $p < 0.005$ (or $T(19) = 2.9$). For hypothesis-guided ROI analysis (i.e., left SFS stimulation site, MNI coordinates: $x = -24, y = 24, z = 36$), we corrected for multiple comparisons using small-volume correction (p < 0.05) restricted within a 10-mm sphere around the target coordinates. We extracted neural betas from this spherical SFS ROI for each participant to perform hypothesis testing and correlational analysis.

## Functional connectivity

We ran a PPI analysis (**Friston et al., 1997**) to investigate the changes in functional connectivity between the left SFS and other brain regions due to cTBS. Here, we extracted physiological time series in the SFS seed region, which corresponds to the time-course of the first eigenvariate across all voxels in the region using principal component analysis (**Friston et al., 1993**). The psychological regressor corresponded to the difference in accumulated evidence, aE (as described in GLM2) between PDM and VDM. We generated PPI estimates from the psychological regressors and the time series from the left SFS, and we then computed the PPI contrasts-of-interest for PDM and VDM. Statistical inference on subject-specific PPI maps was performed using second-level random-effects analysis across participants to allow for group-level inferences. For each participant, we also extracted PPI neural betas, which measure the degree of functional coupling between the left SFS, and we then performed hypothesis testing and correlational analysis.

## Hierarchical Bayesian neural-DDM

We also analysed whether the inclusion of raw trial-by-trial BOLD response extracted from left SFS and attaching it to any of the DDM parameters can improve model evidence. Such a result would suggest that neural activity in the left SFS is directly related to the model's latent decision-relevant parameters. We used $z$-scored single-trial neural beta estimates extracted from the left SFS target site. We implemented four a priori models regarding the role of the left SFS on the decision parameters: Model 1 assumes that the left SFS modulated the decision threshold (**Figure 2—figure supplement 2A**), while Model 2 assumes that left SFS modulated the drift rate (**Figure 2—figure supplement 2B**):

$$\alpha_{c,s,i}^{NEURAL} = \alpha_{c,s,i} + \gamma\theta_{c,s,i},$$

$$\delta_{c,s,i}^{NEURAL} = \kappa E_{c,s,i} + \gamma\theta_{c,s,i},$$

where $\gamma$ is the scale parameter for trial-by-trial left SFS activity, $\theta$. On the other hand, Models 3 and 4 assume that the left SFS modulates both boundary and drift: Model 3 assumes separate scale parameters for each latent process (see *Figure 2—figure supplement 2C*) while Model 4 assumes a common scale parameter for both boundary and drift (see *Figure 2—figure supplement 2D*). Model comparison used the Deviance Information Criterion ($DIC = \bar{D} + pD$), where $pD$ is the effective number of parameters; thus DIC penalises model complexity, and lower DIC denotes better predictive accuracy after accounting for complexity. We then used the best model to re-estimate our latent parameters and to perform Bayesian post hoc inferences.

## Correlating causal changes between neural, latent, and behavioural variables

We tested whether there were any correlational changes between neural, $v$, and behavioural, $\pi$, measures after stimulation. The marginal effect, $r$, measures the correlational change in neural measure, $\nu$, given behavioural measure, $\pi$,

$$r\left(\nu_{c,s} \mid \pi_{c,s}\right) = \frac{\partial}{\partial \pi_{c,s}} \mathbb{E}\left(\nu_{c,s} \mid \pi_{c,s}\right).$$

We test the marginal effect, $r$, of the correlational change between our neural and behavioural measures using our DID regression at trial and subject levels. With trialwise data, we used logit regression to test whether the marginal effect of trialwise changes in left SFS, *Neur*, will affect choices, $\rho$. Similar to previous models, we included various regressors of interest, especially the triple interaction ($Neur \times Task \times TMS$), which accounts for the causal TMS effect, $\phi$, as well as other regressors, $X^k$. The full model is,

$$
\begin{aligned}
\Pr\left(\rho_{(t,c,s,i)}^{DID}\right) = \big(1 \\
+ \exp\big[- \big(\beta_0 + \beta_1 Task_{(t,c,s,i)} + \beta_2 TMS_{(t,c,s,i)} + \beta_3 Neur_{(t,c,s,i)} + \beta_4 \left[Task_{(t,c,s,i)} TMS_{(t,c,s,i)}\right] \\
+ \beta_5 \left[Neur_{(t,c,s,i)} Task_{(t,c,s,i)}\right] + \beta_6 \left[Neur_{(t,c,s,i)} TMS_{(t,c,s,i)}\right] \\
+ \phi \left[Neur_{(t,c,s,i)} Task_{(t,c,s,i)} TMS_{(t,c,s,i)}\right] + \sum_{k=4}^{n} \beta_k X_{(t,c,s,i)}^k \big)\big)^{-1}.
\end{aligned}
$$

To remove nonlinearity bias and isolate the true causal effect, we ran another logit regression without $\phi$,

$$
\begin{aligned}
\Pr\left(\rho_{(t,c,s,i)}^{NODID}\right) = \big(1 \\
+ \exp\big[- \big(\beta_0 + \beta_1 Task_{(t,c,s,i)} + \beta_2 TMS_{(t,c,s,i)} + \beta_3 Neur_{(t,c,s,i)} + \beta_4 \left[Task_{(t,c,s,i)} TMS_{(t,c,s,i)}\right] \\
+ \beta_5 \left[Neur_{(t,c,s,i)} Task_{(t,c,s,i)}\right] + \beta_6 \left[Neur_{(t,c,s,i)} TMS_{(t,c,s,i)}\right] + \sum_{k=4}^{n} \beta_k X_{(t,c,s,i)}^k \big)\big)^{-1},
\end{aligned}
$$

and then we took the difference between the two logit models,

$$\Pr\left(\rho_{(t,c,s,i)}^{TRUEDID}\right) = \Pr\left(\rho_{(t,c,s,i)}^{DID}\right) - \Pr\left(\rho_{(t,c,s,i)}^{NODID}\right).$$

We similarly ran a DID-GLM to test whether the marginal effect of trialwise left SFS neural betas will causally affect RTs,

$$
\begin{aligned}
rt_{(t,c,s,i)} = \beta_0 + \beta_1 Task_{(t,c,s,i)} + \beta_2 TMS_{(t,c,s,i)} + \beta_3 Neur_{(t,c,s,i)} + \beta_4 \left[Task_{(t,c,s,i)} TMS_{(t,c,s,i)}\right] \\
+ \beta_5 \left[Neur_{(t,c,s,i)} Task_{(t,c,s,i)}\right] + \beta_6 \left[Neur_{(t,c,s,i)} TMS_{(t,c,s,i)}\right] \\
+ \phi \left[Neur_{(t,c,s,i)} Task_{(t,c,s,i)} TMS_{(t,c,s,i)}\right] + \varepsilon_{(t,c,s,i)}.
\end{aligned}
$$

We also used cluster-robust standard errors at the subject level in all of our analysis.

With subject-level data, we similarly used linear mixed-effects regression models to test whether the marginal effect of subject-level neural betas (left SFS or PPI) $\nu$, will affect behavioural outcomes or DDM-latent parameters, $\pi$. Similarly, we estimated the marginal effect, $\phi$, with a three-way interaction, $(\nu \times Task \times TMS)$,

$$\pi_{(c,s)} = \beta_0 + \beta_1 Task_{(c,s)} + \beta_2 TMS_{(c,s)} + \beta_3 \nu_{(c,s)} + \beta_4 \left[ Task_{(c,s)} \times TMS \right] + \beta_5 \left[ \nu_{(c,s)} \times Task_{(c,s)} \right]$$
$$+ \beta_6 \left[ \nu_{(c,s)} \times TMS_{(c,s)} \right] + \phi \left[ \nu_{(c,s)} \times Task_{(c,s)} \times TMS_{(c,s)} \right] + \varepsilon_{(c,s)}.$$

This three-way interaction measures whether the correlations between neural activity (left SFS, PPI betas) and behaviour (choice, DDM parameters) are causally affected by stimulation, $TMS$, and whether the effect is specific only during the perceptual task.

## Acknowledgements

This work was supported by grants of the SNSF (105314_152891, CRSII3_141965, and 51NF40_144609) and the SNSF NCCR Affective Sciences to CCR.

## Additional information

### Funding

| Funder | Grant reference number | Author |
| --- | --- | --- |
| Swiss National Science Foundation | 105314_152891 | Christian C Ruff |
| Swiss National Science Foundation | CRSII3_141965 | Christian C Ruff |
| Swiss National Science Foundation | 51NF40_ 144609 | Christian C Ruff |
| Swiss National Science Foundation | NCCR Affective Sciences | Christian C Ruff |

The funders had no role in study design, data collection, and interpretation, or the decision to submit the work for publication.

### Author contributions

Miguel Barretto-Garcia, Conceptualization, Data curation, Formal analysis, Investigation, Visualization, Methodology, Writing – original draft, Project administration, Writing – review and editing; Marcus Grueschow, Conceptualization, Resources, Data curation, Software, Formal analysis, Supervision, Validation, Investigation, Visualization, Methodology, Writing – original draft, Project administration, Writing – review and editing; Marius Moisa, Conceptualization, Resources, Software, Formal analysis, Supervision, Validation, Investigation, Project administration, Writing – review and editing; Rafael Polania, Conceptualization, Resources, Software, Formal analysis, Supervision, Validation, Investigation, Visualization, Methodology, Project administration, Writing – review and editing; Christian C Ruff, Conceptualization, Resources, Software, Supervision, Funding acquisition, Investigation, Methodology, Writing – original draft, Project administration, Writing – review and editing

### Author ORCIDs

Miguel Barretto-Garcia ⬤ https://orcid.org/0000-0001-9054-7859
Marius Moisa ⬤ https://orcid.org/0000-0001-9789-3383
Rafael Polania ⬤ https://orcid.org/0000-0002-6176-6806
Christian C Ruff ⬤ https://orcid.org/0000-0002-3964-2364

### Ethics

Participants completed the MRI screening and consent forms upon their arrival. They then went into the behavioural testing room and read the instructions of the experimental tasks. Participants were fully informed about the study's objectives, the equipment used during the experiment, and MRI safety. We also obtained their consent on the data recorded from them. No participant suffered from any neurological or psychological disorder or took medication that interfered with their participation in the study. Participants received monetary compensation for participation and performance of the perceptual choices, as well as one food item to consume after the experiment

depending on a random value-based choice trial. The experiments conformed to the Declaration of Helsinki and the experimental protocol was approved by the Ethics Committee of the Canton of Zurich.

Reviewer #1 (Public review): https://doi.org/10.7554/eLife.94576.4.sa1
Reviewer #2 (Public review): https://doi.org/10.7554/eLife.94576.4.sa2
Author response https://doi.org/10.7554/eLife.94576.4.sa3

## Additional files

### Supplementary files
Supplementary file 1. Average brain activity that is common for both types of choice (conjunction between PDM and VDM trials). All p-values are FWE-corrected for the whole brain.

Supplementary file 2. Average brain activity that is distinct for both types of choice. All p-values are FWE-corrected for the whole brain. SVC = small-volume correction.

Supplementary file 3. Regions encoding trialwise accumulated evidence (parametric modulation) during perceptual- and value-based decisions, including SFS SVC results for both tasks. Note: Trialwise AE during both types of choices correlated negatively with BOLD activity in intraparietal sulcus (IPS) (peak at $x = -33$, $y = -49$, $z = 58$; *SVC* < 0.05; *Figure 4—figure supplement 1C*) and bilateral fusiform gyrus (right peak at $x = 33$, $y = -49$, $z = -14$; left peak at $x = -30$, $y = -52$, $z = -11$; FWE-corrected with cluster-forming thresholds at $T(19) > 2.9$; *Figure 4—figure supplement 1C*). Note that the inverse of total evidence is directly proportional to the efficiency of evidence accumulation (see **Methods** for more details; SVC = small-volume correction).

Supplementary file 4. Average brain activity that represents efficiency of evidence accumulation for both types of choice (SVC = small-volume correction).

Supplementary file 5. Differences-in-differences results for choice accuracy/consistency and response times. Significance: *p < 0.05, **p < 0.01.

Supplementary file 6. Differences-in-differences results for choice accuracy/consistency and response times. Significance: *p < 0.05, **p < 0.01 (**pre–post cTBS:** Stimulation effect comparing the last two runs during pre-cTBS and the first two runs during post-cTBS; **pre–post cTBS + training:** Stimulation effect comparing all runs during pre-cTBS with the first two runs during post-cTBS; **pre–post cTBS + control variables:** The same as in (a) but we added control variables to test for robustness of the stimulation effect; **pre–post cTBS + training + control variables:** The same as in (b) but we added control variables to test for robustness of the stimulation effect).

Supplementary file 7. Differences-in-differences results for choice accuracy/consistency and response times. Significance: *p < 0.05, **p < 0.01 (**pre–post cTBS:** Stimulation effect comparing the last two runs during pre-cTBS and the first two runs during post-cTBS; **pre–post cTBS + training:** Stimulation effect comparing all runs during pre-cTBS with the first two runs during post-cTBS; **pre–post cTBS + control variables:** The same as in (a) but we added control variables to test for robustness of the stimulation effect; **pre–post cTBS + training + control variables:** The same as in (b) but we added control variables to test for robustness of the stimulation effect).

Supplementary file 8. Hierarchical drift-diffusion model (HDDM) group-level parameter estimates for perceptual decisions (PDM) across TMS conditions and evidence levels ($\delta$ drift, $\alpha$ boundary, $\tau$ non-decision time; DIC reported). Drift values by evidence level summarise the implied $\delta_{c,s,i} = k_{c,s} \times E_{c,s,i}$ for each bin; separate drift parameters were not estimated for each evidence level.

Supplementary file 9. Hierarchical drift-diffusion model (HDDM) group-level parameter estimates for value-based decisions (VDM) across TMS conditions and evidence levels ($\delta$, $\alpha$, $\tau$; DIC reported). As in *Supplementary file 8*, evidence-level $\delta$ values are descriptive summaries of $\delta_{c,s,i} = k_{c,s} \times E_{c,s,i}$ at each evidence level, not independently fitted drift parameters.

Supplementary file 10. Hierarchical drift-diffusion model (HDDM) participant-level parameter estimates for PDM ($\delta$, $\alpha$, $\tau$) with model fit (DIC) per subject.

Supplementary file 11. Hierarchical drift-diffusion model (HDDM) participant-level parameter estimates for VDM ($\delta$, $\alpha$, $\tau$) with model fit (DIC) per subject.

MDAR checklist

## Data availability

Behavioural and neuroimaging data and the code for data analysis are available at https://doi.org/10.17605/OSF.IO/3DMH9.

The following dataset was generated:

| Author(s) | Year | Dataset title | Dataset URL | Database and Identifier |
|-----------|------|---------------|-------------|-------------------------|
| Barretto-Garcia M, Grueschow M, Polania R, Moisa M, Ruff CC | 2023 | Data and Code for Causal evidence for a domain-specific role of left superior frontal sulcus in perceptual decision making | https://doi.org/10.17605/OSF.IO/3DMH9 | Open Science Framework, 10.17605/OSF.IO/3DMH9 |

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

## Appendix 1

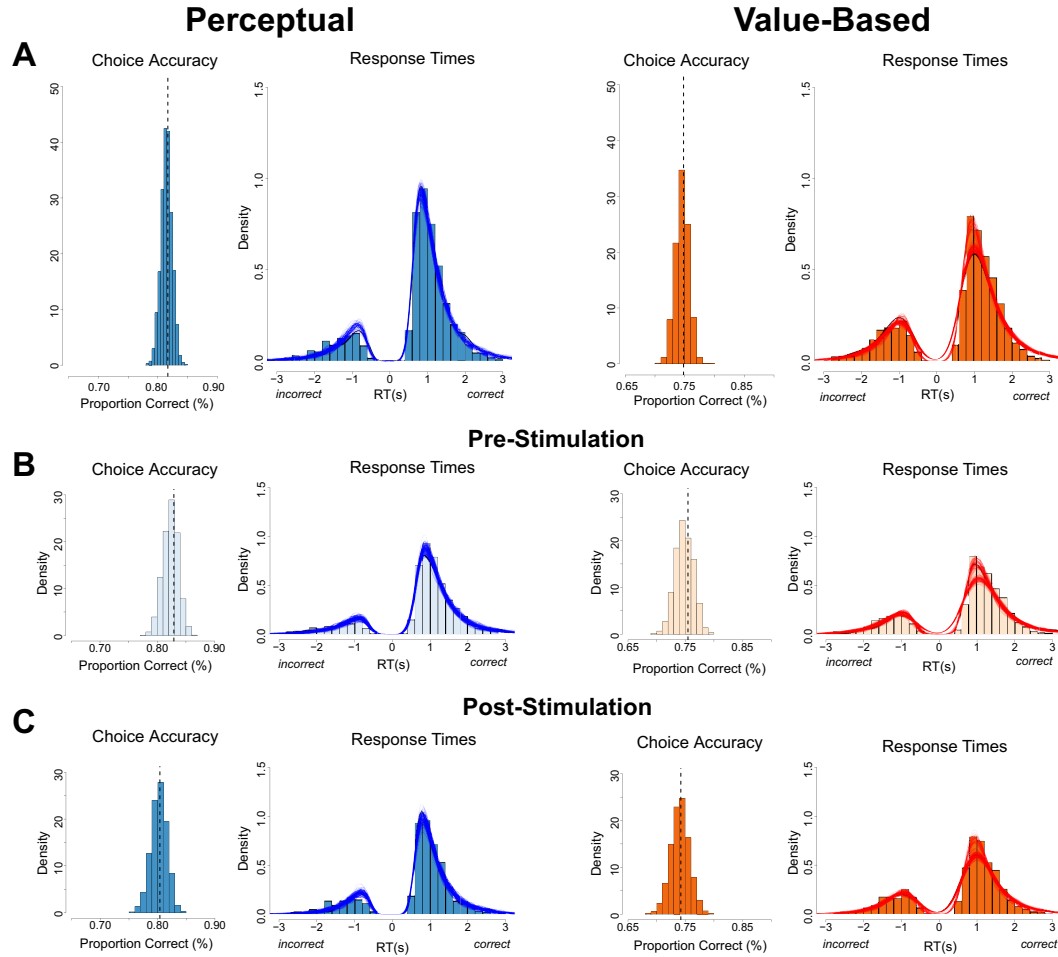

**Appendix 1—figure 1.** Posterior-predictive checks: overall accuracy and RT distributions. Posterior-predictive simulations from the fitted hierarchical drift-diffusion model (HDDM) compared to observed data. Left column: histograms show the simulated distribution of mean accuracy across 3000 posterior draws; the vertical dashed line marks the observed mean accuracy. Right column: observed RT histograms (positive = correct; negative = error) with posterior-predictive density curves overlaid. Panels show (**A**) PDM versus VDM (pooled over TMS), (**B**) pre-TMS, and (**C**) post-TMS. The model reproduces both accuracy and RT patterns in each condition.

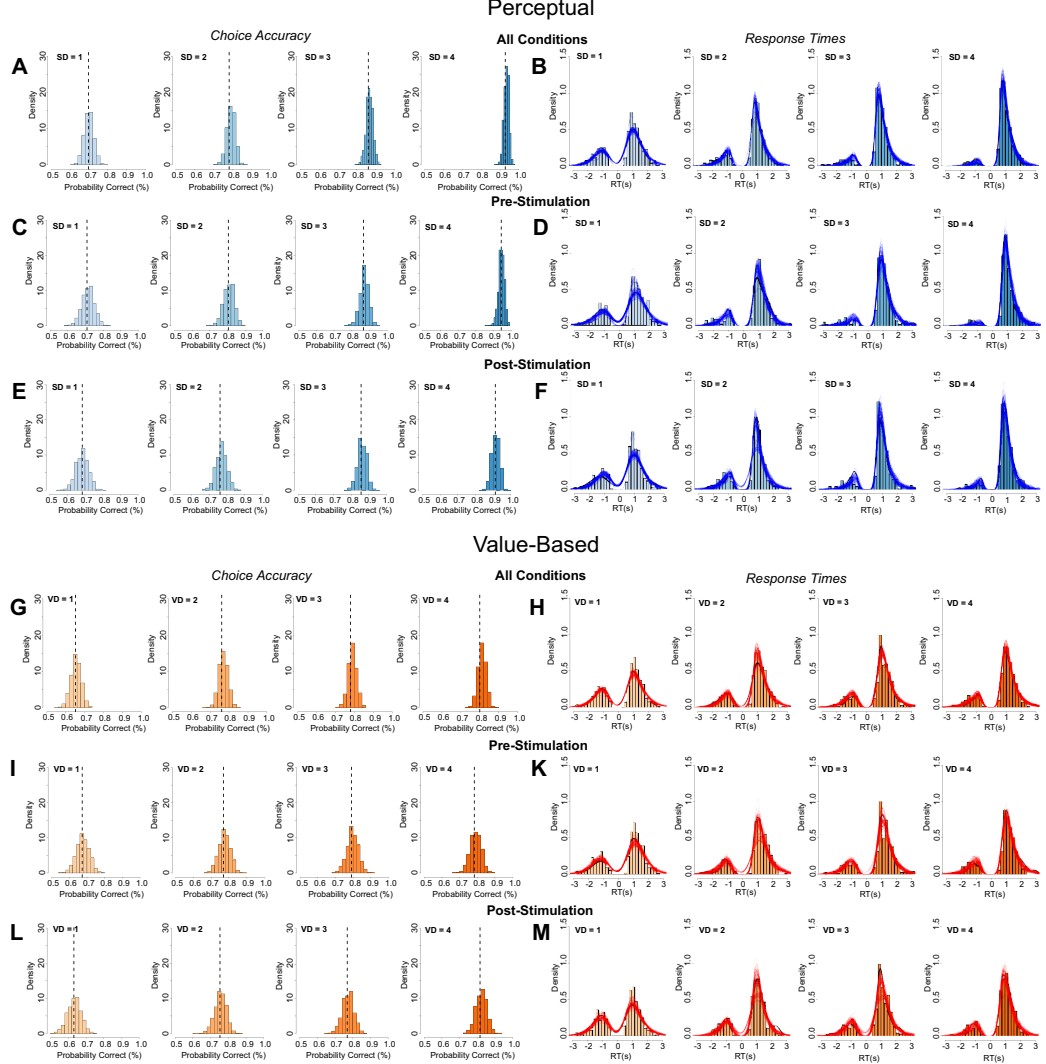

**Appendix 1—figure 2.** Posterior-predictive checks by evidence level. Similar to *Appendix 1—figure 1*, but split by evidence levels 1–4 (left-to-right within each row). Rows display: (top) PDM pooled over TMS for (**A**) choice accuracy and response times (**B**), (middle) pre-TMS (**C, D**), and (bottom) post-TMS (**E, F**); the corresponding three rows for VDM are arranged analogously for pooled (**G, H**), pre-TMS (**I, K**) and post-TMS (**L, M**). Simulations closely match accuracy and RT distributions at each evidence level in both tasks.

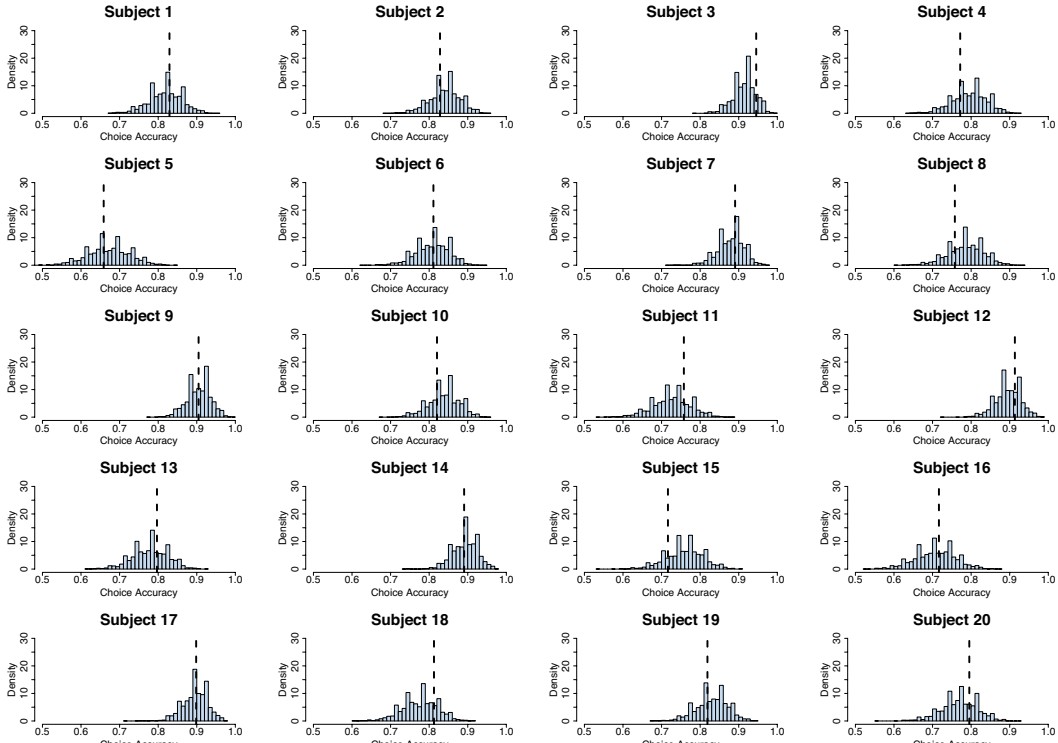

**Appendix 1—figure 3.** Subject-level accuracy fits for PDM. For each participant, the histogram shows the posterior-predictive distribution of that participant's mean accuracy from 3000 simulations; the vertical dashed line marks the observed mean accuracy for that participant. The hierarchical drift-diffusion model (HDDM) captures the dispersion of accuracies across individuals in PDM.

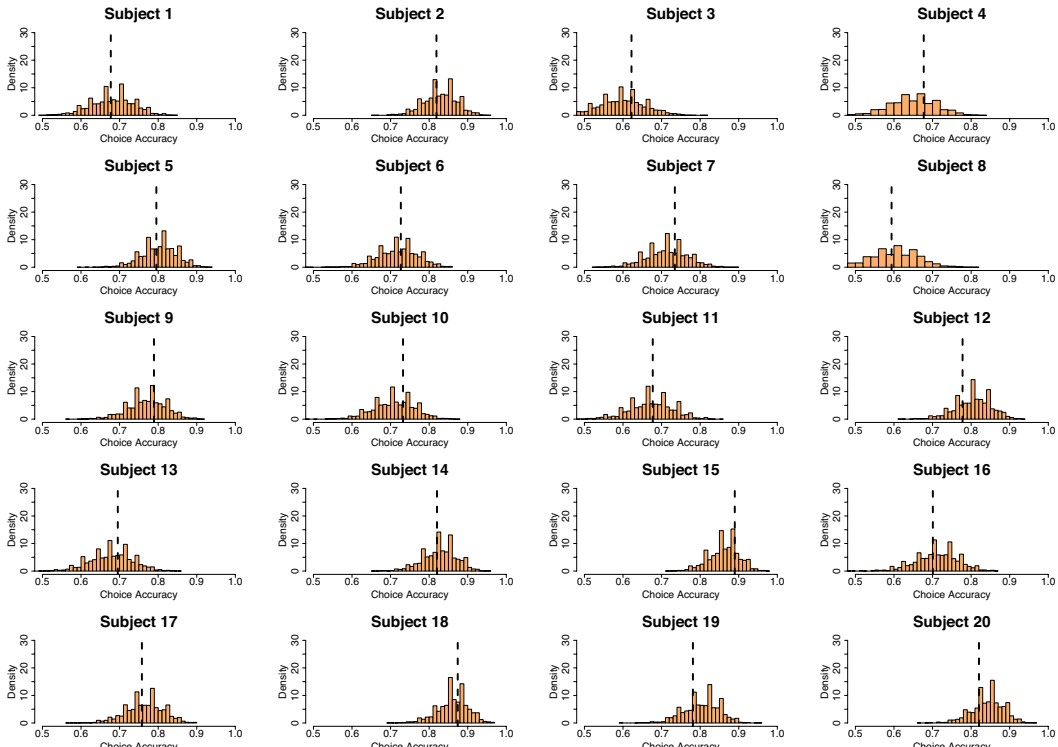

**Appendix 1—figure 4.** Subject-level accuracy fits for VDM. Same format as *Appendix 1—figure 3*, for VDM. Posterior-predictive distributions align with observed subject-level accuracies, indicating good recovery of between-subject variability in the value-based task.

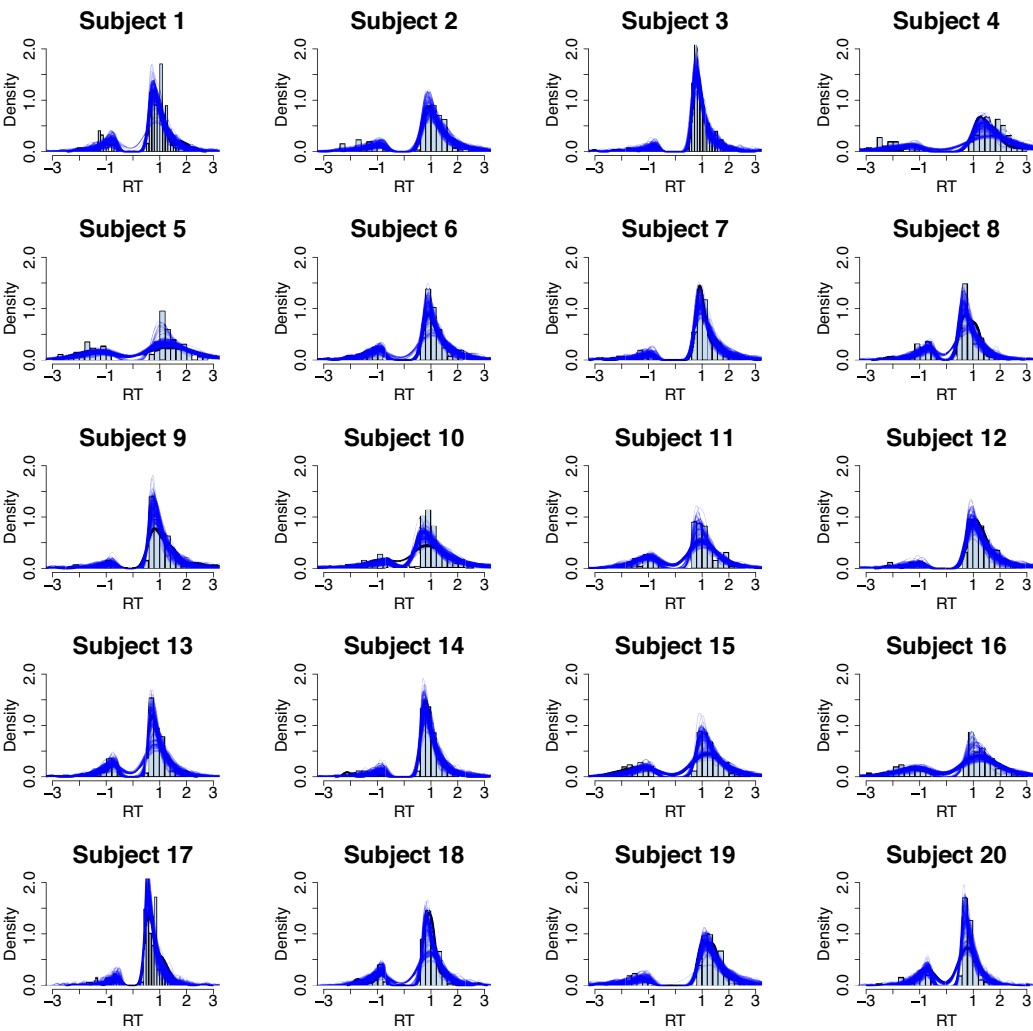

**Appendix 1—figure 5.** Subject-level RT distribution fits for PDM. For each participant, observed PDM RT histograms (positive = correct; negative = error) are overlaid with posterior-predictive density curves from the hierarchical drift-diffusion model (HDDM). The model reproduces the full shape of individual RT distributions, including correct/error asymmetries.

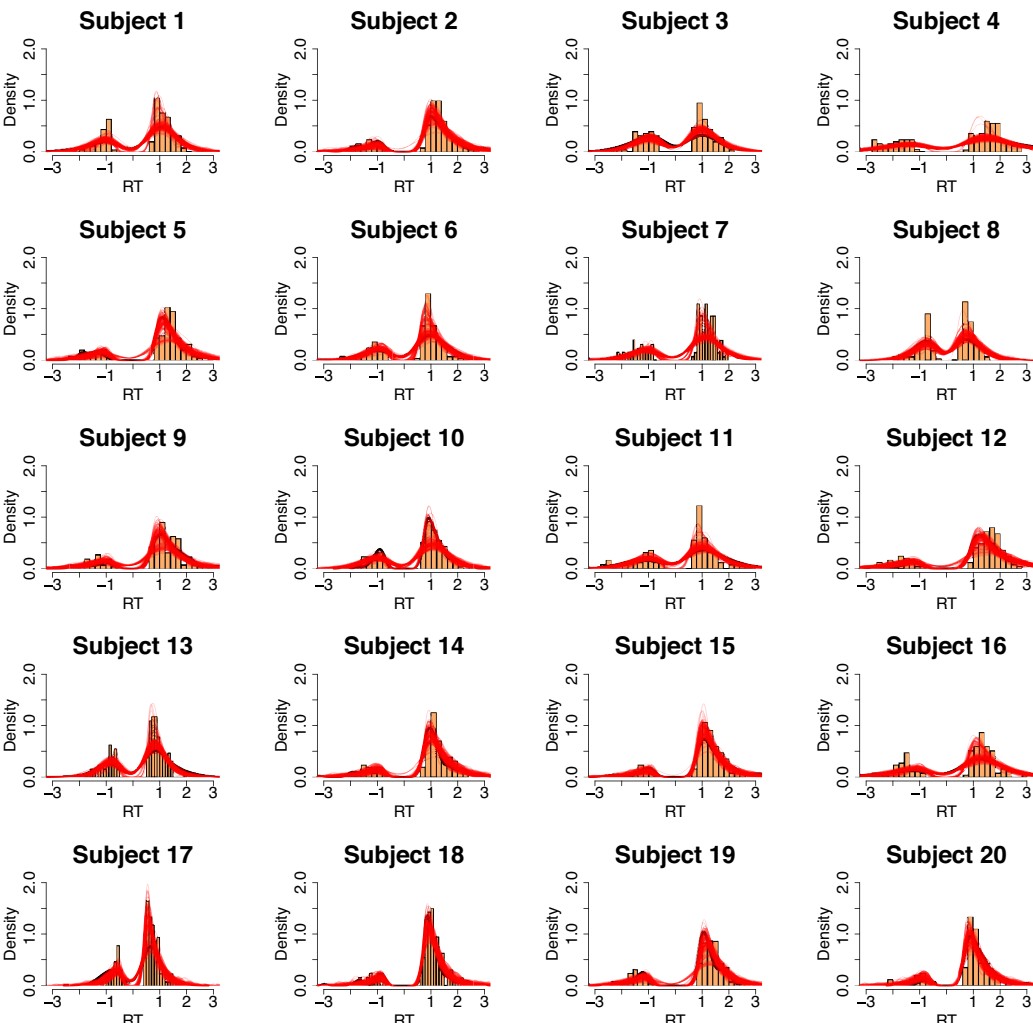

**Appendix 1—figure 6.** Subject-level RT distribution fits for VDM. Same format as *Appendix 1—figure 5*, for VDM. Posterior-predictive densities closely track the observed RT distributions across participants.

