## [Editor Report · eLife Assessment]

In this **important** paper, Garcia et al seek to determine whether the superior frontal sulcus (SFS), an area previously implicated in evidence accumulation for perceptual decisions, plays a causal role in perceptual and/or value-based decisions. Through a combination of careful paradigm design, computational modelling, transcranial magnetic stimulation and fMRI analyses, the authors provide **convincing** evidence that the SFS supports perceptual but not value-based decisions and that its disruption leads to a lowering of decision boundaries.

---

## [Referee Report · Reviewer #1 (Public review)]

Summary:

In this study, participants completed two different tasks. A perceptual choice task in which they compared the sizes of pairs of items and a value-different task in which they identified the higher value option among pairs of items with the two tasks involving the same stimuli. Based on previous fMRI research, the authors sought to determine whether the superior frontal sulcus (SFS) is involved in both perceptual and value-based decisions or just one or the other. Initial fMRI analyses were devised to isolate brain regions that were activated for both types of choices and also regions that were unique to each. Transcranial magnetic stimulation was applied to the SFS in between fMRI sessions and it was found to lead to a significant decrease in accuracy and RT on the perceptual choice task but only a decrease in RT on the value-different task. Hierarchical drift diffusion modelling of the data indicated that the TMS had led to a lowering of decision boundaries in the perceptual task and a lowering of non-decision times on the value-based task. Additional analyses show that SFS covaries with model derived estimates of cumulative evidence, that this relationship is weakened by TMS.

The paper has many strengths including the rigorous multi-pronged approach of causal manipulation, fMRI and computational modelling which offers a fresh perspective on the neural drivers of decision making. Some additional strengths include the careful paradigm design which ensured that the two types of tasks were matched for their perceptual content while orthogonalizing trial-to-trial variations in choice difficulty. The paper also lays out a number of specific hypotheses at the outset regarding the behavioural outcomes that are tied to decision model parameters and well justified.

---

## [Referee Report · Reviewer #2 (Public review)]

Summary:

The authors set out to test whether a TMS-induced reduction in excitability of the left Superior Frontal Sulcus influenced evidence integration in perceptual and value-based decisions. They directly compared behaviour-including fits to a computational decision process model---and fMRI pre and post TMS in one of each type of decision-making task. Their goal was to test domain-specific theories of the prefrontal cortex by examining whether the proposed role of the SFS in evidence integration was selective for perceptual but not value-based evidence.

Strengths:

The paper presents multiple credible sources of evidence for the role of the left SFS in perceptual decision making, finding similar mechanisms to prior literature and a nuanced discussion of where they diverge from prior findings. The value-based and perceptual decision making tasks were carefully matched in terms of stimulus display and motor response, making their comparison credible.

---

## [Author Response]

The following is the authors’ response to the previous reviews

**Public Reviews:**

**Reviewer #1 (Public review):**
Summary:In this study, participants completed two different tasks. A perceptual choice task in which they compared the sizes of pairs of items and a value-different task in which they identified the higher value option among pairs of items with the two tasks involving the same stimuli. Based on previous fMRI research, the authors sought to determine whether the superior frontal sulcus (SFS) is involved in both perceptual and value-based decisions or just one or the other. Initial fMRI analyses were devised to isolate brain regions that were activated for both types of choices and also regions that were unique to each. Transcranial magnetic stimulation was applied to the SFS in between fMRI sessions and it was found to lead to a significant decrease in accuracy and RT on the perceptual choice task but only a decrease in RT on the value-different task. Hierarchical drift diffusion modelling of the data indicated that the TMS had led to a lowering of decision boundaries in the perceptual task and a lower of nondecision times on the value-based task. Additional analyses show that SFS covaries with model derived estimates of cumulative evidence, that this relationship is weakened by TMS.Strengths:The paper has many strengths, including the rigorous multi-pronged approach of causal manipulation, fMRI and computational modelling, which offers a fresh perspective on the neural drivers of decision making. Some additional strengths include the careful paradigm design, which ensured that the two types of tasks were matched for their perceptual content while orthogonalizing trial-to-trial variations in choice difficulty. The paper also lays out a number of specific hypotheses at the outset regarding the behavioural outcomes that are tied to decision model parameters and well justified.

We thank the reviewer for their thoughtful summary of the study and for highlighting these strengths. We are pleased that the multi-pronged approach combining causal manipulation, fMRI, and hierarchical drift–diffusion modelling, as well as the careful matching of perceptual content across the two tasks, came across clearly. We also appreciate the reviewer’s positive remarks on the specificity of our a priori hypotheses and their links to decision-model parameters. In revising the manuscript, we have aimed to further streamline the presentation of these hypotheses and to more explicitly connect the behavioural predictions, model parameters, and neural readouts throughout the Results and Discussion sections.

Weaknesses:In my previous comments (1.3.1 and 1.3.2) I noted that key results could be potentially explained by cTBS leading to faster perceptual decision making in both the perceptual and value-based tasks. The authors responded that if this were the case then we would expect either a reduction in NDT in both tasks or a reduction in decision boundaries in both tasks (whereas they observed a lowering of boundaries in the perceptual task and a shortening of NDT in the value task). I disagree with this statement. First, it is important to note that the perceptual decision that must be completed before the value-based choice process can even be initiated (i.e. the identification of the two stimuli) is no less trivial than that involved in the perceptual choice task (comparison of stimulus size). Given that the perceptual choice must be completed before the value comparison can begin, it would be expected that the model would capture any variations in RT due to the perceptual choice in the NDT parameter and not as the authors suggest in the bound or drift rate parameters since they are designed to account for the strength and final quantity of value evidence specifically. If, in fact, cTBS causes a general lowering of decision boundaries for perceptual decisions (and hence speeding of RTs) then it would be predicted that this would manifest as a short NDT in the value task model, which is what the authors see.

We thank the reviewer for raising these points and for the helpful clarification. We agree that, in principle, the architecture of the value-based task can be conceived as involving an upstream perceptual process that must be completed, to some degree, before value comparison can proceed. Under such a multistage framework, it is indeed possible that cTBS-induced changes in a perceptual decision stage could manifest as a reduction in boundary separation in the *pure* perceptual task, while the same perturbation appears as a shortening of non-decision time (NDT) when fitting a single-stage DDM to the value task. In this sense, our earlier statement that a “general speeding effect” would necessarily produce identical parameter changes (either NDT or boundaries) in both tasks was too strong, and we are grateful to the reviewer for pointing this out.

At the same time, this alternative explanation remains fully compatible with our central claim that the left SFS plays a perceptual rather than value-based role. We agree with the reviewer that there must be a stimulus-related circuit (in visual and parietal regions) that encodes the physical attributes of the options, and that this upstream processing can influence both tasks. However, a large body of work suggests that left SFS is not part of this primary identification circuitry, but rather contributes specifically to the accumulation and comparison of sensory evidence (e.g., Heekeren et al., 2004, 2006), downstream from areas such as FFA, PPA, or MT/V5 that encode stimulus identity. In other words, stimulus identification (forming a representation of “what is where”) is anatomically and functionally distinct from the accumulation of evidence toward a perceptual decision. Within this framework, the reviewer’s proposal that cTBS speeds “perceptual decisions” across tasks can be understood as targeting precisely the evidence-accumulation stage we ascribe to SFS, with the value-comparison stage proper likely implemented in other regions (e.g., vmPFC and connected valuation circuitry).

We therefore do not rely solely on the dissociation between boundary changes in the perceptual task and NDT changes in the value task as decisive evidence against a “general speeding” account. Instead, our interpretation is based on the convergence of behavioural, model-based, and neural results. First, in the perceptual task, cTBS to left SFS leads to a selective reduction in decision boundary and a concomitant change in trialwise BOLD activity within the stimulated region that covaries with perceptual choice behaviour and with the latent decision variable inferred from the HDDM. Second, in the value task, cTBS does not affect value sensitivity or accuracy, nor does it alter value-related drift or boundary parameters; the only robust HDDM effect is a modest shortening of NDT. Third, critically, left SFS BOLD activity is modulated by perceptual evidence and by cTBS in the perceptual task, but we observe no evidence that SFS activity encodes value evidence or shows value-related cTBS neuronal effects in the value task.

Taken together, these findings indicate that the left SFS serves a causal role in the accumulation of perceptual evidence and in the setting of the choice criterion for perceptual decisions. The reviewer’s suggestion that cTBS may induce a general speeding of perceptual processes that also influences the value task is compatible with this conclusion, in the sense that any contribution of SFS to the value task is best understood as acting via a perceptual component that is upstream of value comparison, rather than via the value accumulation process itself. We have clarified this point in the Discussion of the revised manuscript and now explicitly acknowledge that our DDM dissociation alone does not exclude a general perceptual speeding account, but that the combination of task-specific neural effects in SFS, preserved value-based choice behaviour, and the absence of value-related BOLD changes in SFS strongly support a primarily perceptual role for this region.

**Reviewer #2 (Public review):**
Summary:The authors set out to test whether a TMS-induced reduction in excitability of the left Superior Frontal Sulcus influenced evidence integration in perceptual and value-based decisions. They directly compared behaviour-including fits to a computational decision process model---and fMRI pre and post TMS in one of each type of decision-making task. Their goal was to test domain-specific theories of the prefrontal cortex by examining whether the proposed role of the SFS in evidence integration was selective for perceptual but not value-based evidence.Strengths:The paper presents multiple credible sources of evidence for the role of the left SFS in perceptual decision making, finding similar mechanisms to prior literature and a nuanced discussion of where they diverge from prior findings. The value-based and perceptual decision-making tasks were carefully matched in terms of stimulus display and motor response, making their comparison credible.

We thank the reviewer for their clear summary of our aims and approach, and for highlighting these strengths. We are pleased that the convergence between causal TMS, fMRI, and hierarchical modelling comes across as providing credible evidence for the role of left SFS in perceptual decision-making, and that our attempt to link these results to the existing literature is seen as appropriately nuanced. We also appreciate the reviewer’s positive assessment of the task design, in particular the close matching of perceptual content and motor output across perceptual and value-based decisions, which was central to our goal of testing domain-specific theories of prefrontal function. In revising the manuscript, we have further clarified these design choices and their rationale, and we have streamlined the exposition of how the hypotheses, model parameters, and neural readouts are connected across the two decision domains.

Weaknesses:I was confused about the model specification in terms of the relationship between evidence level and drift rate. While the methods (and e.g. supplementary figure 3) specify a linear relationship between evidence level and drift rate, suggesting, unless I misunderstood, that only a single drift rate parameter (kappa) is fit. However, the drift rate parameter estimates in the supplementary tables (and response to reviewers) do not scale linearly with evidence level.

We thank the reviewer for raising this point and appreciate the opportunity to clarify the model specification. In our hierarchical DDM, we did *not* fit separate, free drift parameters for each evidence level. As shown in Supplementary Fig. 3, the drift on each trial is specified as\begin{document}$$\displaystyle \delta_{c, s, i}=\kappa_{c, s} \times E_{c, s, i}$$\end{document}

where 𝐸_𝑐,𝑠,𝑖_ the trial-wise evidence (difference in size or value) and κ_𝑐,𝑠_ is a single drift-scaling parameter per condition and session. Thus, the linear dependence of drift on evidence is implemented at the *trial level* via 𝜅; we do not estimate independent 𝛿 parameters for each evidence level.

In Supplementary Tables 8 and 9 we report, for descriptive purposes, the posterior means of 𝛿 conditional on each evidence bin (levels 1–4), alongside the corresponding decision boundary and nondecision time summaries. These values are therefore *derived* quantities that reflect the combination of (i) the single κ_𝑐,𝑠_ parameter, (ii) the empirical distribution of continuous evidence values 𝐸 within each bin, and (iii) hierarchical pooling across subjects and sessions. Consequently, they are expected to increase monotonically with evidence level—as they do in our data—but not to lie exactly on a straight line in the discrete level index, because the underlying evidence bins are not equally spaced in physical units and because of between-subject variability and posterior uncertainty.

We will revise the text and table captions to make clear that the evidence-level entries are descriptive summaries of 𝛿 implied by the 𝜅×𝐸 formulation, rather than independently estimated drift parameters, in order to avoid this confusion.

-The fit quality for the value-based decision task is not as good as that for the PDM, and this would be worth commenting on in the paper.

We agree that the HDDM fit for the value-based task is somewhat weaker than for the perceptual task. This is reflected in the somewhat higher DIC values for VDM compared with PDM and in slightly broader posterior-predictive distributions (Supplementary Tables 8–11 and Supplementary Figs. 11–16). We believe this difference primarily reflects the greater intrinsic variability of subjective value-based choices (e.g. trial-to-trial fluctuations in preferences, satiety, or attention), coupled with our decision to use the same relatively simple DDM architecture for both tasks to allow a principled cross-task comparison. Importantly, posterior-predictive checks show that, for VDM as well, the model adequately reproduces both accuracy and full RT distributions at the group and subject level (Supplementary Figs. 11–16), indicating that the fit quality is sufficient for our purposes. In the revised manuscript we now explicitly note that the model captures PDM behaviour more tightly than VDM and that this may reduce sensitivity to very small cTBS effects on value-based decision parameters, even though no systematic effects are evident in our data. Crucially, our central conclusion—that left SFS plays a domain-specific role in setting the decision boundary for perceptual evidence—relies on the robust behavioural, computational, and neural effects observed in PDM and does not depend on assuming a perfect model fit for VDM.

- Supplementary Figure 3 specifies the distribution for kappa hyper-parameter twice.

We thank the reviewer for spotting this typo. We have revised Supplementary Figure 3 legend.